# Dynamic Sparsity Is Channel-Level Sparsity Learner

**Lu Yin[1] , Gen Li[2], Meng Fang[3], Li Shen[4], Tianjin Huang[1], Zhangyang Wang[5],**
**Vlado Menkovski[1]**, **Xiaolong Ma[2]**, **Mykola Pechenizkiy[1]**, **Shiwei Liu[1,5]**
[1]Eindhoven University of Technology, [2]Clemson University
[3]University of Liverpool, [4]JD Explore Academy, [5]University of Texas at Austin,
{l.yin,t.huang,m.pechenizkiy,v.menkovski,s.liu3}@tue.nl
gen@g.clemson.edu, xiaolom@clemson.edu, Meng.Fang@liverpool.ac.uk
mathshenli@gmail.com, atlaswang@utexas.edu

## Abstract

Sparse training has received an upsurging interest in machine learning due to its tantalizing saving potential for the entire training process as well as inference. Dynamic sparse training (DST), as a leading sparse training approach, can train deep neural networks at high sparsity from scratch to match the performance of their dense counterparts. However, most if not all DST prior arts demonstrate their effectiveness on unstructured sparsity with highly irregular sparse patterns, which receives limited support in common hardware. This limitation hinders the usage of DST in practice. In this paper, we propose **Ch**annel-**a**ware dynamic **s**pars**e** (**Chase**), which for the first time seamlessly translates the promise of unstructured dynamic sparsity to GPU-friendly channel-level sparsity (not fine-grained *N:M* or group sparsity) during one end-to-end training process, without any ad-hoc operations. The resulting small sparse networks can be directly accelerated by commodity hardware, without using any particularly sparsity-aware hardware accelerators. This appealing outcome is partially motivated by a hidden phenomenon of dynamic sparsity: *off-the-shelf unstructured DST implicitly involves biased parameter reallocation across channels, with a large fraction of channels (up to 60%) being sparser than others.* By progressively identifying and removing these channels during training, our approach translates unstructured sparsity to channel-wise sparsity. Our experimental results demonstrate that Chase achieves **1.7×** inference throughput speedup on common GPU devices without compromising accuracy with ResNet-50 on ImageNet. We release our codes in https://github.com/luuyin/chase.

## 1 Introduction

Deep neural networks (DNNs) have recently demonstrated impressive breakthroughs with increasing scales [2; 8; 36]. Besides the well-known scaling, i.e., test accuracy scales as a power law regarding model size and training data size in quantity [18; 26], recent work has observed that massive increases in quantity can imbue models with qualitatively new behavior [49]. However, the memory and computation required to train and deploy these large models can be a heavy burden on the environment and finance [13; 43]. Therefore, people start to probe the possibility of training sparse neural networks from scratch without involving any dense training steps (dubbed sparse training [39; 35]). As the memory requirement and multiplications (which dominate neural network computation) associated with zero weights can be skipped, sparse training is becoming a promising direction due to their "end-to-end" saving potentials for both efficient training and efficient inference.

Sparse training can be categorized into two groups, static sparse training and dynamic sparse training according to the dynamics of the sparse pattern during training. Static sparse training

37th Conference on Neural Information Processing Systems (NeurIPS 2023).

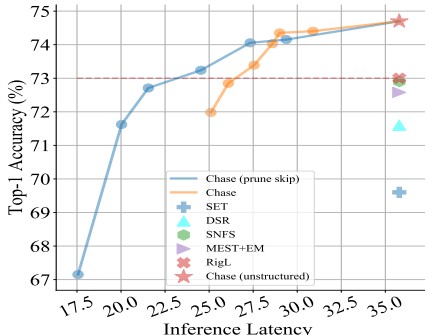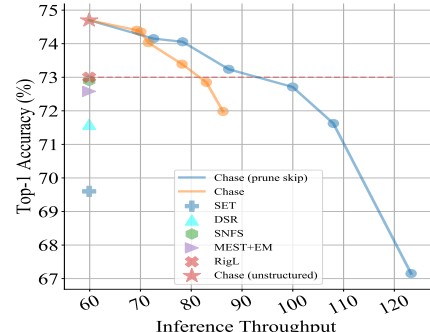

Figure 1: Inference latency and throughput of various DST methods. The sparsity level is 90% for all approaches. All models are trained for 100 epochs with the ResNet-50/ImageNet benchmark. Each dot of Chase and Chase (prune skip) corresponds to a model with distinct channel-wise sparsity. The results of latency are obtained on NVIDIA 2080TI GPU with a batch size of 2.

(SST) [28; 56; 53; 33; 22], namely, draws a sparse neural network at initialization before training and train with a fixed sparse pattern (sparse connections between layers) without further changes. Dynamic sparse training (DST) [39; 10; 35], on the contrary, jointly optimizes the weights and sparse patterns during training, usually delivering better performance than the static ones. DST quickly evolves as a leading direction in sparse training due to its compelling performance and training/inference efficiency. For instance, a sparse ResNet-34 with only 2% parameters left can be dynamically trained to match the performance of its dense counterpart without involving any pre-training or dense training [35].

While showing promise in performance and efficiency, so far the real speedup of DST has only been demonstrated on CPU [34; 6] or IPU [6]. Most sparse training methods produce unstructured sparse neural networks with extremely irregular sparse patterns, which can not be directly accelerated in common hardware (i.e., GPU and TPU), compared to the straightforward and hardware-friendly sparse pattern produced by channel pruning [17; 37].

Many endeavors strive to solve this issue by coarsening the sparsity granularity, which can be loosely categorized into two groups. i. Grouping nonzero weights into blocks. As GPU performs very fast on contiguous memory operations, block-wise sparsity enjoys much more speedups than unstructured sparsity in practice. Group lasso regularization [41; 14] is a widely-used technique to induce block sparsity in the network. Ad-hoc grouping operations can also be utilized to build dense blocks from unstructured sparse weights [46; 3]. ii. Seeking fine-grained structured sparse patterns. For instance, inspired by the recent support of 2:4 fine-grained sparsity in NVIDIA Ampere [42], previous arts attempt to find a sweet spot between structured and unstructured sparsity by learning *N:M* sparsity patterns [60; 20; 45]. However, these methods either rely on specialized sparse-aware accelerators [9; 42] to enable speedups or suffer from significant performance degradation due to the constraint location of nonzero values [25].

In this paper, we propose a new method dubbed **Ch**annel-**a**ware dynamic **s**parse (**Chase**), which can effectively transfer the promise of unstructured sparse training into the hardware-friendly channel sparsity with comparable or even better performance on common GPU devices. The roadmap of our exploration is as follows:

- ■ **Observation 1:** We first present an emerging characteristic of DST: off-the-shelf DST approaches implicitly involve biased parameter reallocation, resulting in a large proportion of channels (up to 60%) that rapidly become sparser than their initializations at the very early training stage. We term them as "sparse amenable channels" for the sake of convenient reference.

- ■ **Observation 2:** We examine the prunability (i.e., the accuracy drop caused by pruning) of the sparse amenable channels, we find that these channels cause marginal damages to the model performance than their counterparts when pruned.

- **A New Metric:** We propose a new, sparsity-inspired, channel pruning metric – Unmasked Mean Magnitude (UMM) – that can be used to precisely discover sparse amenable channels during training by monitoring the quantity and quality of weight sparsity.

- **A New Approach:** Based on the above findings, we propose **Ch**annel-**a**ware dynamic **spars**e (**Chase**), a first sparse training framework that can favorably transform unstructured sparsity into channel-wise sparsity on the fly. Chase starts with an unstructured sparse neural network and dynamically trains it while gradually eliminating sparse amenable channels with the lowest UMM scores. During training, we globally grow and shrink parameters to strengthen performance further.

- **Performance:** Chase inherently can be tailed into an unstructured sparse training approach and a structured sparse training approach. Our unstructured variant establishes a new state-of-the-art accuracy bar for sparse training. More impressively, our structured approach is able to maintain or even surpass SoTA performance with ResNet-50 on ImageNet, while achieving $1.2\times$ - $1.7\times$ inference throughput speedups on common GPU devices.

## 2    Sparse Amenable Channels in DST

We first describe the basis and notations of the prior sparse training arts. Afterward, we provide evidence for the existence of the sparse amenable channels during the dynamic sparse training across different architectures and demonstrate that pruning of such channels leads to marginal performance damage than their counterparts. Based on this interesting finding, we introduce Chase, a sparsity-inspired sparse training method that for the first time translates the theoretical promise of sparse training into GPU-friendly speedup, without using any specialized CUDA implementations.

### 2.1    Prior Sparse Training Arts

Let us denote the sparse neural network as $f(\boldsymbol{x}; \boldsymbol{\theta}_{\mathrm{s}})$. $\boldsymbol{\theta}_{\mathrm{s}}$ refers to a subset of the full network parameters $\boldsymbol{\theta}$ at a sparsity level of $(1 - \frac{\|\boldsymbol{\theta}_{\mathrm{s}}\|_0}{\|\boldsymbol{\theta}\|_0})$ and $\| \cdot \|_0$ represents the $\ell_0$-norm.

It is common to initialize sparse subnetworks $\boldsymbol{\theta}_{\mathrm{s}}$ randomly based on the uniform [40; 5] or non-uniform layer-wise sparsity ratios with *Erdős-Rényi* (ER) graph [39; 10; 35; 31]. In the case of image classification, sparse training aims to optimize: $\hat{\boldsymbol{\theta}}_{\mathrm{s}} = \operatorname{argmin}_{\boldsymbol{\theta}_{\mathrm{s}}} \sum_{i=1}^{\mathrm{N}} \mathcal{L}(f(x_i; \boldsymbol{\theta}_{\mathrm{s}}), y_i)$ using data $\{(x_i, y_i)\}_{i=1}^{\mathrm{N}}$, where $\mathcal{L}$ is the loss function. Static sparse training (SST) maintains the same sparse network connectivity during training after initialization. Dynamic sparse training (DST), on the contrary, allows the sparse subnetworks to dynamically explore new parameters while sticking to a fixed sparsity budget. Most of the DST methods follow a simple prune-and-grow scheme [39] to perform parameter exploration, i.e., pruning $r$ proportion of the least important parameters based on their magnitude, and immediately grow the same number of parameters randomly [39] or using the potential gradient [10]. Formally, the parameter exploration can be formalized as the following two steps:

$$\boldsymbol{\theta}_{\mathrm{s}} = \Psi(\boldsymbol{\theta}_{\mathrm{s}}, \ r), \tag{1}$$

$$\boldsymbol{\theta}_{\mathrm{s}} = \boldsymbol{\theta}_{\mathrm{s}} \cup \Phi(\boldsymbol{\theta}_{i \notin \boldsymbol{\theta}_{\mathrm{s}}}, \ r). \tag{2}$$

where $\Psi$ is the specific pruning criterion and $\Phi$ is growing scheme. These metrics may vary from sparse training method to another. In addition to prune-and-grow, previous work [23; 47] dynamically activates top-K parameters during forward-pass while keeping a larger number of parameters updated in backward-pass to get rid of dense calculation of gradient. At the end of the training, sparse training can converge to a performant sparse subnetwork. Since the sparse neural networks are trained from scratch, the memory requirements and training/inference FLOPs are only a fraction of their dense counterparts.

One daunting drawback of sparse training is the resulting subnetworks are usually imbued with extremely irregular sparsity patterns, therefore, receiving very limited support from common hardware like GPU and TPU.

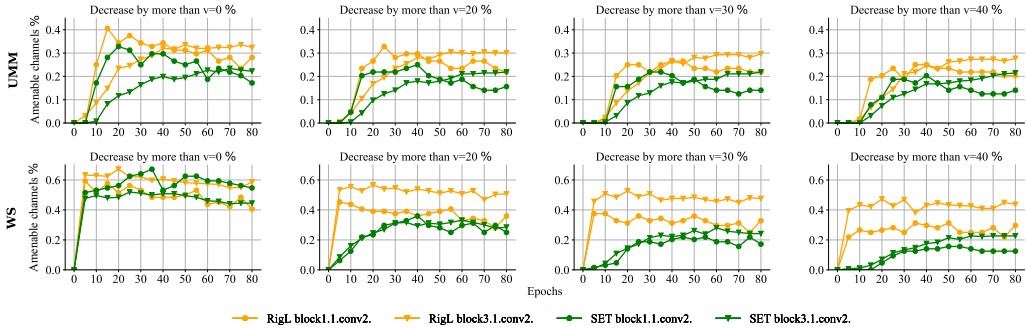

Figure 2: The portion of sparse amenable channels justified by two metrics, the Unmasked Mean Magnitude (UMM) and the Weight Sparsity (WS), of ResNet-50 trained on CIFAR-100.

## 2.2 Sparse Amenable Channels

Here, we introduce the most important cornerstone concept for this work - "sparse amenable channels" - which is defined as the channels whose sparsity becomes higher than their initial values caused by dynamic sparse training.

To provide empirical evidence for this interesting observation, we visualize the training dynamics of DST by monitoring two specific metrics of channels, Weight Sparsity and Unmasked Weight Magnitude, which are defined below.

**Weight Sparsity (WS)** (Quantity): Weight Sparsity directly quantizes the emergence of the Sparse Amenable Channels in quantity. Larger weight sparsity means more elements in the channel are becoming zero. Consequently, channels with fewer non-zero weights than their initial status are justified as Sparse Amenable Channels in this case.

**Unmasked Mean Magnitude (UMM)** (Quantity and Quality): Instead of solely quantitatively monitoring the weight sparsity, it is preferable to take the quality (i.e., magnitude) of the nonzero weights into consideration due to the crucial role of magnitude to dynamic sparse training [39; 10; 35]. Here, Unmasked Mean Magnitude refers to the mean magnitude of all the weights (including zero and nonzero) in the channel without considering masking. Smaller Unmasked Mean Magnitude represents the channels that come to be more sparse both in quantity and quality. Specifically, channels with fewer non-zero parameters but larger magnitudes will be excluded from the Sparse Amenable Channels. Therefore, the number of Sparse Amenable Channels justified here will be smaller than WS. We formalize these two metrics in Table 1 for a better interpretability. For comparison, we also evaluate the Masked Mean Magnitude (MMM), i.e., the mean magnitude of the non-zero weights.

We determined channels at the $i$ training iteration are amenable if their values of Weight Sparsity are larger than their initialized values by a ratio $v$ or their values of Unmasked Mean Magnitude are smaller than their initialized values by a ratio $v$: $\frac{\text{WS}_i - \text{WS}_0}{\text{WS}_0} > v$ or $\frac{\text{UMM}_0 - \text{UMM}_i}{\text{UMM}_0} > v$. In other words, we say a channel becomes $v$ more sparse than its initial status if its $\text{WS}_i$ surpasses $\text{WS}_0$ by $v$, or its $\text{UMM}_i$ is smaller than $\text{UMM}_0$ by $v$.

Taking the most representative DST approaches SET [39] and RigL [10] as examples, we measure the number of the Sparse

Table 1: **Metrics that are introduced to measure the dynamics of the Sparse Amenable Channels.** The weight tensor and the binary mask of a channel is represented with $\boldsymbol{\theta}$ and $\boldsymbol{m}$, respectively. And $\|\cdot\|_0$ stands for the $\ell_0$-norm.

| | |
|---|---|
| Weight Sparsity (WS) | $1 - \frac{\|\boldsymbol{m}\odot\boldsymbol{\theta}\|_0}{\|\boldsymbol{\theta}\|_0}$ |
| Unmasked Mean Magnitude (UMM) | $\frac{\sum\|\boldsymbol{\theta}\|}{\|\boldsymbol{\theta}\|_0}$ |
| Masked Mean Magnitude (MMM) | $\frac{\sum\|\boldsymbol{m}\odot\boldsymbol{\theta}\|}{\|\boldsymbol{m}\odot\boldsymbol{\theta}\|_0}$ |

Amenable Channels across layers in Figure 2, with $v$ equals 0%, 20%, 30%, and 40%. We summarize our main observations here. ❶ Overall, we observe that a large part of channels (up to 60%) tend to be sparse amenable. While the number of amenable channels tends to decrease as $v$ increases, there still exists around 10% ∼ 40% amenable channels becoming 40% more sparse than their initializations

**Algorithm 1:** Pseudocode of Chase

**Input:** Sparse neural network with initialized sparse weight $\boldsymbol{\theta}_s$, target sparsity $s_p$, current sparsity $s_t$, parameter update frequency $\Delta T_p$, sparsity mutation factor $s_e$, target channel sparsity $S_c$, channel pruning frequency $\Delta T$, current channel sparsity $S_t$, training steps $\tau$, exploration stop steps $\tau_{stop}$, total training steps $\tau_{total}$

**Output:** A sparse model satisfying the target sparsity $s_p$ and channel sparsity requirement $S_c$.

**while** $\tau < \tau_{stop}$ **do**

    **if** $Mod(t, \Delta T) = 0$ *and* $t \leq T_{max}$ **then**

        $\boldsymbol{\theta}_s \leftarrow$ `Global channel prune`$(\boldsymbol{\theta}_s, S_t)$ ▷ *Perform gradual amenable channel pruning using Eq. 3*

    **if** $Mod(t, \Delta T_p) = 0$ *and* $t \leq T_{max}$ **then**

        $\boldsymbol{\theta}_s(s_t = s_p - s_e) \leftarrow$ `Global Parameter Grow`$(\boldsymbol{\theta}_s)$

        Training for $\Delta\tau$ epochs, $\tau \leftarrow \tau + \Delta T_p$

        $\boldsymbol{\theta}_s(s_t = s_p) \leftarrow$ `Global Parameter Prune`$(\boldsymbol{\theta}_s)$    ▷ *Chase adopts global parameter exploration and soft memory bound.*

Continue sparse training from the epoch $\tau_{stop}$ to $\tau_{end}$.

---

across layers. ❷ Fewer channels are justified as sparse amenable channels using the UMM metric than WS, as we expected. ❸ Deeper layers suffer from more amenable channels than shallow layers. ❹ RigL tends to extract more amenable channels than SET at the very early training phase. A possible reason is that the dense gradient encourages RigL to quickly discover and fill weights to the important (non-amenable) channels compared to the random growth used in SET.

**Sparse amenable channels enjoy better prunability[1] than their counterparts.** So far, we have unveiled the existence of the sparse amenable channels. It is natural to conjecture that these amenable channels can be a good indicator for channel pruning. To evaluate our conjecture, we choose the above proposed two metrics, Weight Sparsity (WS) and Unmasked Mean Magnitude (UMM), as our pruning criteria and perform a simple one-shot global channel pruning after regular DST training in comparison with their reversed metrics as well as several commonly-used principles, including random pruning [33], network slimming [37], and Masked Mean Magnitude (MMM). Channels with the highest values

Table 2: Top-1 test accuracy (%) of various channel pruning criteria with ResNet-50 on CIFAR-100. "Reverse" refers to pruning with the reversed metric.

| Method | Channel Sparsity | | |
|---|---|---|---|
| | 10% | 20% | 30% |
| Standard RigL [10] | 76.89±0.43 | 76.89±0.43 | 76.89±0.43 |
| Random Pruning [33] | 43.01±9.62 | 11.74±2.79 | 3.79±1.32 |
| Network Slimming [37] | 76.82±0.43 | 76.67±0.39 | 66.57±2.95 |
| MMM | 62.31±8.66 | 19.34±14.88 | 5.32±2.88 |
| MMM Reverse | 5.28±2.52 | 2.04±0.30 | 1.72±0.40 |
| WS | 76.86±0.43 | 76.79±0.39 | 62.79±5.42 |
| WS Reverse | 2.9±0.91 | 2.43±0.07 | 2.03±0.38 |
| UMM | **76.88±0.43** | **76.90±0.42** | **71.77±2.31** |
| UMM Reverse | 3.18±0.48 | 2.23±0.26 | 1.51±0.35 |

are pruned for WS, and the ones with the smallest values are pruned for UMM. Table 2 shows that both WS and UMM achieve good performance and UMM performs the best. Meanwhile, their reversed metrics perform no better than random pruning. Perhaps more interestingly, the resulting hybrid channel-level sparse models favorably preserve the performance of the unstructured RigL with no accuracy drop when pruned with mild channel sparsity.

In addition, we also observe the existence of "sparse amenable channel" in a broad range of settings, including ResNet-32/VGG-16 on CIFAR-100, MLP Model on CIFAR10, and ViT Small, ResNet-50 on ImageNet in Appendix. Hence, we believe that sparse amenable channels is a very general phenomenon that widely exists across different architectures and datasets.

This encouraging result confirms our conjecture and demonstrates the promising potentials of sparse amenable channels (UMM) as a strong metric to remove channels during training. In the next section, we will explain in detail how we leverage Sparsity Amenable Channels and UMM to translate the promise of unstructured sparse training to the hardware-friendly sparse neural networks.

---

[1]Prunability here refers to the accuracy drop caused by the channel removal.

# 3 Methodology - Chase

Inspired by the above encouraging findings of sparse amenable channels, we introduce **Ch**ase-**a**ware dynamic **s**pars**e** (**Chase**) in this section. We follow the widely-used sparse training framework used in [39; 10]. The technical novelty of Chase mainly lies in two aspects. On the structured sparsity level, we adopt the gradual sparsification schedule [61] to gradually remove Amenable Channels during training with smallest UMM scores. The gradual sparsification schedule provides us with a moderate sparsification schedule, favorably relieving the accuracy drop caused by aggressive channel pruning. On the unstructured sparsity level, we globally redistribute parameters based on their magnitude and gradient, which significantly strengthens the sparse training performance. The overall Pseudocode of Chase is illustrated in Algorithm 1. We provide technical details of the above components below.

**Gradual Amenable Channel Pruning.** The gradual sparsification schedule is widely-used in the unstructured sparse literature to produce strong unstructured sparse subnetworks [61; 11; 31]. We explore it to the channel pruning regime with several ad-hoc modifications. Let us denote the initial and target final channel-wise sparsity level as $S_i$ and $S_f$, respectively; gradual pruning starts at the training step $t_0$ with pruning frequency $\Delta T$, performing over a span of $n$ pruning steps. The sparsity level $S_t$ at pruning step $t$ is:

$$S_t = S_f + (S_i - S_f) \left( 1 - \frac{t - t_0}{n\Delta T} \right)^3.$$  (3)

We globally collect UMM (see Section 2.2 for the definition) of each channel as the pruning criterion and progressively remove the sparse amenable channels with the smallest UMM according to Eq 3. We observe that layer collapse occurs sometimes without setting layer-wise pruning constraints. To avoid layer collapse, we use $\beta$ to control the minimum number of channels remaining in layer $l$ to be $(1 - S_f) \cdot \beta \cdot w_l$, where $w_l$ is the number of channels in layer $l$. We empirically find that smaller $\beta$ tends to yield better performance. We report more details Appendix A.2.

To maintain the overall number of parameters the same during training, we redistribute the overly pruned parameters back to the remaining channels at the next parameter grow phase using Eq 2. We find that without doing this will significantly hurt the model's performance.

**Global Parameter Exploration.** Global parameter exploration was introduced in previous arts [40; 5]. However, with the popularity of RigL [10], it is common to use a fixed set of layer-wise sparsities. Here, we revisit global parameter exploration in DST. To be specific, we globally prune parameters that have the smallest magnitudes and grow parameters with highest gradient magnitude. This small adaption brings a large performance benefit to RigL (up to 2.62% on CIFAR-100 and 1.7% on ImageNet), reported as "Chase ($S_c = 0$)" in Table 3 and Table 4.

**Soft Memory Bound.** Soft memory bound was proposed in [59], which allows the parameter growing operation happens before the parameter pruning, improving the performance at the cost of a slight increase of memory requirements and FLOPs. We borrow the idea of soft memory bound to allow parameters firstly being added to the existing parameters followed by $\Delta T_p$ iteration of training, then remove the less important parameters including the newly added ones. This can avoid forcing the existing weights in the model to be removed if they are more important than newly grown weights.

After training, Chase slims down the initial "big sparse" model to a "small sparse" model with a significantly reduced number of channels. We completely remove the pruned channels in the current layer as well as the corresponding input dimensions of the next layer, so that the produced small sparse models can directly enjoy the acceleration in GPU.

# 4 Experimental Evaluation of Chase

In this section, we comprehensively evaluate Chase in comparison with the various state-of-the-art (SOTA) unstructured sparse training methods as well as the state-of-the-art channel-pruning algorithms. At last, we provide a detailed analysis of hyperparameters and perform an ablation study to evaluate the effectiveness of the components of Chase.

Our evaluation is conducted with two widely used model architectures VGG-19 [48] and ResNet-50 [15] on across various datasets including CIFAR-10/100 and ImageNet, We summarize the

Table 3: Test accuracy (%) of the sparse VGG-19 and ResNet-20/50 on CIFAR-10/100.

| Dataset | CIFAR-10 | | | CIFAR-100 | | |
|---|---|---|---|---|---|---|
| Sparsity | 90% | 95% | 98% | 90% | 95% | 98% |
| **VGG-19** (Dense) | 93.85±0.05 | 93.85±0.05 | 93.85±0.05 | 73.43±0.08 | 73.43±0.08 | 73.43±0.08 |
| SynFlow [53] | 93.35 | 93.45 | 92.24 | 71.77 | 71.72 | 70.94 |
| GraSP [56] | 93.30 | 93.04 | 92.19 | 71.95 | 71.23 | 68.90 |
| SNIP [28] | 93.63 | 93.43 | - | 72.84 | 71.83 | - |
| Chase+GraSP ($S_c = 0.5$) | 94.06±0.22 | 93.88±0.06 | **93.89±0.20** | 73.17±0.09 | 72.81±0.11 | **71.66±0.15** |
| Chase+SNIP ($S_c = 0.5$) | **94.83±0.06** | **95.08±0.14** | - | **78.26±0.26** | **77.16±0.04** | - |
| Deep-R [1] | 90.81 | 89.59 | 86.77 | 66.83 | 63.46 | 59.58 |
| SET [39] | 93.61±0.13 | 93.09±0.25 | 91.81±0.04 | 72.58±0.12 | 71.48±0.12 | 69.04±0.15 |
| RigL [10] | 93.60±0.09 | 93.05±0.06 | 91.95±0.15 | 72.92±0.31 | 71.85±0.53 | 69.57±0.24 |
| MEST [59] | 93.61±0.36 | 93.46±0.41 | 92.30±0.44 | 72.52±0.37 | 71.21±0.41 | 69.02±0.34 |
| Chase ($S_c = 0$) | 94.02±0.13 | **93.89±0.12** | 93.60±0.05 | **73.54±0.12** | **73.05±0.25** | **72.19±0.33** |
| Chase ($S_c = 0.5$) | **94.03±0.11** | 93.84±0.08 | **93.69±0.03** | 73.43±0.12 | 73.04±0.22 | 71.85±0.18 |
| **ResNet-50** (Dense) | 94.75±0.01 | 94.75±0.01 | 94.75±0.01 | 78.23±0.18 | 78.23±0.18 | 78.23±0.18 |
| SynFlow [53] | 92.49 | 91.22 | 88.82 | 73.37 | 70.37 | 62.17 |
| SNIP [28] | 92.65 | 90.86 | - | 73.14 | 69.25 | - |
| GraSP [56] | 92.47 | 91.32 | 88.77 | 73.28 | 70.29 | 62.12 |
| Chase+SNIP ($S_c = 0.5$) | 93.99±0.09 | 93.89±0.10 | - | 73.44±0.02 | 72.80±0.05 | - |
| Chase+GraSP ($S_c = 0.5$) | **94.78±0.35** | **94.71±0.07** | **94.36±0.15** | **77.70±0.24** | **77.65±0.22** | **75.74±0.24** |
| Deep-R [1] | 91.62 | 89.84 | 86.45 | 66.78 | 63.90 | 58.47 |
| SET [39] | 94.65±0.01 | 94.05±0.06 | 92.98±0.18 | 76.14±0.54 | 75.90±0.19 | 73.21±0.06 |
| RigL [10] | 94.42±0.17 | 94.22±0.23 | 93.20±0.08 | 77.18±0.42 | 76.50±0.26 | 74.84±0.13 |
| Chase ($S_c = 0$) | **94.95±0.02** | **94.87±0.02** | 94.15±0.17 | **78.11±0.11** | **78.14±0.28** | 76.88±0.31 |
| Chase ($S_c = 0.5$) | 94.88±0.03 | 94.85±0.18 | **94.20±0.18** | 77.52±0.30 | 77.48±0.62 | **77.03±0.29** |
| **ResNet-20** (Dense) | 92.55±0.02 | 92.55±0.02 | 92.55±0.02 | 68.65±0.19 | 68.65±0.19 | 68.65±0.19 |
| SNIP [28] | 88.06±0.07 | 84.21±0.33 | 74.61±0.40 | 54.40±0.09 | 42.45±0.65 | 24.55±0.56 |
| GraSP [56] | 88.35±0.12 | 84.95±0.30 | 78.25±0.22 | 55.49±0.08 | 45.96±0.15 | 30.67±0.94 |
| SET [39] | 90.16±0.09 | 87.70±0.09 | 83.41±0.04 | 62.08±0.16 | 54.77±0.74 | 43.70±0.92 |
| RigL [10] | 89.82±0.10 | 87.44±0.33 | 79.16±0.96 | 60.49±0.17 | 52.97±0.23 | 31.94±1.52 |
| Chase ($S_c = 0$) | **90.43±0.16** | **88.65±0.29** | **85.26±0.29** | **62.18±0.05** | **57.38±0.41** | **47.06±0.65** |
| Chase ($S_c = 0.5$) | 89.98±0.45 | 88.65±0.02 | 85.24±0.18 | 60.88±0.19 | 55.78±0.37 | 46.91±0.41 |

implementation details for Chase in Appendix B. To show the superior performance of Chase on unstructured and structured sparsity, we report two variants of Chase: Chase ($S_c = 0$) represents the unstructured version without channel pruning and Chase ($S_c = 0.5$) stands for the structured version with 50% channel-level sparsity.

## 4.1 Comparison with off-the-shelf SOTA DST

**CIFAR-10/100.** Our method is naturally versatile and can be applied to both static sparse training and dynamic sparse training regimes. Therefore, for each model, we categorize the results into two groups and report the results in Table 3.

❶ Chase dramatically improves the performance of SST. As shown in the upper panel of each group, Chase significantly boosts the accuracy (up to 13.62% for GraSP with ResNet-50 on CIFAR-100) of SST methods like SNIP and GraSP. ❷ Chase ($S_c = 0$) establishes a new state-of-the-art performance bar for unstructured sparse training. Compared with the SOTA DST methods such as RigL and MEST, we clearly see that Chase ($S_c = 0$) universally outperforms all the presented DST methods by a large margin. Specifically, Chase ($S_c = 0$) achieves 3.67 % and 3.15% performance gains compared with SET on ResNet-50 and VGG-19. We also notice that the performance gain on CIFAR-100 is larger than the ones on CIFAR-10, which is as expected since CIFAR-100 has a larger improvement space than CIFAR-10. ❸ Chase ($S_c = 0.5$), with only 50% channels remaining, matches the performance of its unstructured variant, demonstrating the promise of the unstructured DST can be favorably transferred to the structured regime.

**ImageNet**. We reported the results on ImageNet in Table 4. Again, Chase ($S_c = 0$) dominates the performance in the unstructured sparsity regime, achieving 1.7% and 2.12% accuracy improvements over RigL and MEST, respectively. While the accuracy of Chase ($S_c = 0.4$) slightly decreases by 0.67% compared to Chase ($S_c = 0$), it still outperforms RigL by a good margin (1.03%), while enjoying a $1.5\times$ real inference speedup on common GPU.

Table 4: Test accuracy (%) of sparse ResNet-50 on ImageNet trained with 100 epochs and 150 epochs ($1.5\times$). Training FLOPs of sparse training methods are normalized with the FLOPs used to train a dense model. "GPU-supported FLOPs" refers to the real FLOPs that are required to calculate on a common GPU which usually does not support irregular sparsity patterns.

| Method | Top-1 Accuracy | Theoretical FLOPs (Train) | Theoretical FLOPs (Test) | GPU-Supported FLOPs (Test) | TOP-1 Accuary | Theoretical FLOPs (Train) | Theoretical FLOP (Test) | GPU-Supported FLOPs (Test) |
|---|---|---|---|---|---|---|---|---|
| ResNet-50 (Dense) | 76.8±0.09 | 1x (3.2e18) | 1x (8.2e9) | 1x (8.2e9) | 76.8±0.09 | 1x (3.2e18) | 1x (8.2e9) | 1x (8.2e9) |
| Sparsity | | | 80% | | | | 90% | |
| SET [39] | 72.9±0.39 | 0.23× | 0.23× | 1.00× | 69.6±0.23 | 0.10× | 0.10× | 1.00× |
| DSR [40] | 73.3 | 0.40× | 0.40× | 1.00× | 71.6 | 0.30× | 0.30× | 1.00× |
| SNFS [5] | 75.2±0.11 | 0.61× | 0.42× | 1.00× | 72.9±0.06 | 0.50× | 0.24× | 1.00× |
| RigL [10] | 75.1±0.05 | 0.42× | 0.42× | 1.00× | 73.0±0.04 | 0.25× | 0.24× | 1.00× |
| MEST [59] | 75.39 | 0.23× | 0.21× | 1.00× | 72.58 | 0.12× | 0.11× | 1.00× |
| RigL-ITOP [35] | 75.84±0.05 | 0.42× | 0.42× | 1.00× | 73.82±0.08 | 0.25× | 0.24× | 1.00× |
| Chase ($S_c = 0$) | **75.87** | 0.37× | 0.34× | 1.00× | **74.70** | 0.24× | 0.21× | 1.00× |
| Chase ($S_c = 0.3$) | 75.62 | 0.39× | 0.36× | 0.75× | 74.35 | 0.25× | 0.22× | 0.74× |
| Chase ($S_c = 0.4$) | 75.27 | 0.39× | 0.37× | 0.68× | 74.03 | 0.26× | 0.23× | 0.67× |
| MEST$_{1.5\times}$ | 75.73 | 0.40× | 0.21× | 1.00× | 75.00 | 0.20× | 0.11× | 1.00× |
| Chase$_{1.5\times}$ ($S_c = 0$) | **76.67** | 0.55× | 0.34× | 1.00× | **75.77** | 0.36× | 0.21× | 1.00× |
| Chase$_{1.5\times}$ ($S_c = 0.3$) | 76.23 | 0.57× | 0.36× | 0.75× | 75.20 | 0.37× | 0.22× | 0.74× |
| Chase$_{1.5\times}$ ($S_c = 0.4$) | 76.00 | 0.59× | 0.37× | 0.68× | 74.87 | 0.38× | 0.23× | 0.67× |

When increasing the training time to $1.5\times$ (150 epochs), Chase also demonstrates a promising scaling trend. Chase ($S_c = 0$) matches the dense performance with only $0.55\times$ training FLOPs and consistently outperforms the off-the-shelf best DST method, MEST. Again, the promising results of Chase ($S_c = 0$) can be effectively translated to channel-level sparsity. Chase ($S_c = 0.4$) is $1.5\times$ faster than MEST, while still performing on par or even better than MEST.

**Real Inference Speedups.** We further compare the actual inference throughput and latency of our model against RigL in Table 5. All results are averaged from 100 individual runs with one NVIDIA 2080TI GPU in float32 on PyTorch. We set the batch size to 128 for CIFAR-100 and 2 for ImageNet, when evaluating the latency. We empirically find that pruning the skip connection leads to a significant accuracy drop while providing benefits on speedups. Therefore, the standard Chase keeps the skip connection layers untouched for optimal accuracy. To fully unleash Chase's potential on real inference speedups, we also provide another variant of Chase that prunes skip connection layers, dubbed Chase (prune skip). Compared with SoTA RigL, Chase (prune skip) is able to prune 50% channels with ResNet-50 on ImageNet, leading to a notable 68% of throughput gain, while only losing 0.29% accuracy. Even without pruning skip connections, our model is about to provide 31% throughput speedups, while outperforming RigL by 0.39%.

Table 5: Inference throughput and latency. The best results are marked in bold.

| Method | Dataset | Model | $s_p$ | $S_c$ | Accuracy (%) (↑) | Throughput (↑) | Latency (ms) (↓) |
|---|---|---|---|---|---|---|---|
| RigL [10] | CIFAR-100 | VGG-19 | 0.9 | 0.0 | 72.92±0.31 | 15274.31 | 8.42 |
| Chase | CIFAR-100 | VGG-19 | 0.9 | 0.5 | **73.43±0.12** | **24981.77 (64%↑)** | **5.37 (36%↓)** |
| RigL [10] | CIFAR-100 | ResNet-50 | 0.9 | 0.0 | 77.18±0.42 | 3095.13 | 44.23 |
| Chase | CIFAR-100 | ResNet-50 | 0.9 | 0.5 | **77.52±0.30** | **3958.67 (28%↑)** | **34.76 (21%↓)** |
| RigL [10] | ImageNet | ResNet-50 | 0.9 | 0.0 | 73.00±0.04 | 59.50 | 35.79 |
| Chase | ImageNet | ResNet-50 | 0.9 | 0.5 | 73.39±0.04 | 78.19 (31%↑) | 27.55 (23%↓) |
| Chase (prune skip) | ImageNet | ResNet-50 | 0.9 | 0.5 | 72.71±0.03 | **99.97 (68%↑)** | **21.55 (40%↓)** |
| Chase | ImageNet | ResNet-50 | 0.9 | 0.4 | **74.03±0.03** | 71.54 (20%↑) | 30.13 (16%↓) |
| Chase (prune skip) | ImageNet | ResNet-50 | 0.9 | 0.4 | 73.24±0.03 | 87.38 (47%↑) | 24.53 (31%↓) |

## 4.2 Extensive Analysis

**Performance under different channel sparsity.** To investigate the performance of different channel sparsity with the same parameter count, we maintain 2%, 5% parameters and prune the model to different channel sparsity ranging from 20% to 70%. The results are reported in Figure 3. The same training scheme is adopted as Section 4.1. Two DST baselines, RigL and SET are adopted for comparison. Not surprisingly, the more channels remain the better performance of the model archives. Notably, in all settings, Chase archives better performance than the baselines with just 50% channels, and Chase outperforms SET and RigL with just 30% channels on ResNet-50 at 98% sparsity.

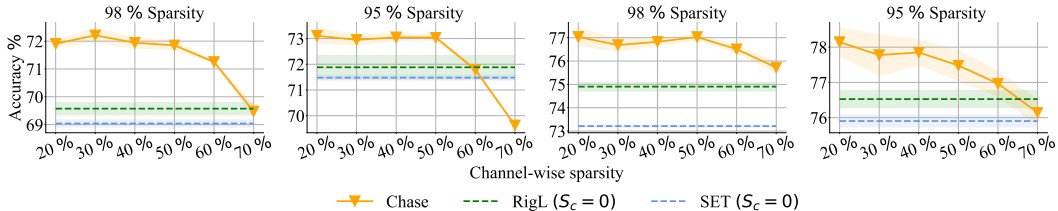

Figure 3: Performance of Chase under different channel sparsity. For Rigl and SET, we keep the channels un-pruned as baselines.

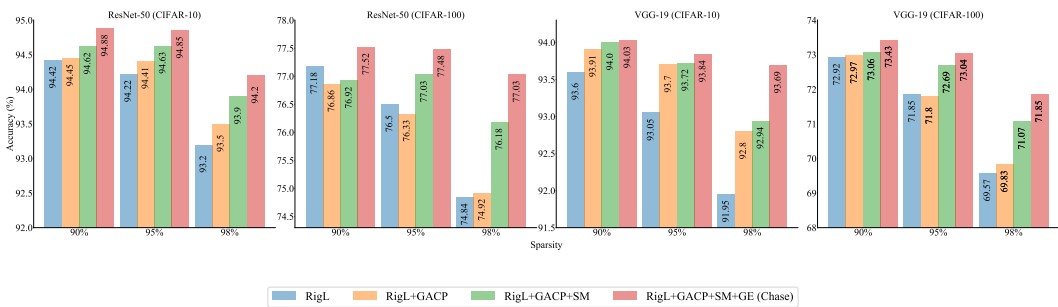

Figure 4: Ablation Study of Chase. GACP denotes gradual amenable channel pruning (50% channel sparsity), SM indicates soft memory bound, GE represents global parameter exploration.

**Effect of the channel pruning frequency.** We also study how the channel pruning frequency $\Delta T$ affects Chase's performance. For all experiments, we fixed the ending time $\tau_{stop}$ for gradual amenable channel pruning as 130 epochs, the total training epochs $\tau_{total}$ as 160 epochs and the minimum channel ratio factor as $\beta$ as 0.5, while altering $\Delta T$ to 1000, 4000, 8000, and 16000 iterations. We report the results in Appendix A.4. Overall, the largest $\Delta T$ 16000 leads to worse performance. This observation is as expected, as we aim to achieve the same channel sparsity and larger $\Delta T$ results in more removed channels in each punning operation. Consequently, larger performance degradation will be introduced during each pruning which could degrade the training stability.

**Ablation study.** In Figure 4, we study the effectiveness of different components in Chase, namely, the soft memory constraint (SM) and global parameter exploration (GE) on CIFAR-10/100 with ResNet-50 and VGG-19. We denote the RigL as our baseline, as RigL applies magnitude-based pruning and gradients-based growth like Chase. We apply the same training recipe as described in Section 4.1. Gradually amenable channel pruning safely removes 50% channels from RigL, while only suffering from minor or even no performance degradation. As for SM and GE, we found these techniques all bring universal performance improvements. Surprisingly, adding SM results in a 1.26% accuracy increase on CIFAR-100 with ResNet-50 at 98% sparsity. With GE, we can obtain a more optimal layer-wise ratio, which also consistently improves the accuracy from SM.

## 5  Related Work

Recently, as the financial and environmental costs of model training grow exponentially [50; 43], endeavors start to pursue training efficiency by investigating training sparse neural networks from scratch. Most Sparse training works can be divided into two categories, static sparse training, and dynamic sparse training. Static sparse training determines the structure of the sparse network at the initial stage of training by using certain pre-defined layer-wise sparsity ratios [38; 39; 10; 33].

Dynamic sparse training is designed to reduce the computation as well as memory footprint during the whole training phase. It trains a sparse neural network from scratch while allowing the sparse mask to be updated during training. SET [39] update sparse mask at the end of each training epoch by magnitude-based pruning and random growing. DSR [40] develops a dynamic reparameterization method that allows parameter reallocation during dynamic mask updating. DeepR [1] combines dynamic sparse parameterization with stochastic parameter updates for training. RigL [10] and

SNFS [5] propose to uses gradient information to grow weights. ITOP [35] studies the underlying mechanism of DST and discovers that the benefits of DST come from searching across time all possible parameters. GraNet [31] introduces the concept of "pruning plasticity" and quantitatively studies the effect of pruning throughout training. MEST [59] proposes a memory-friendly training framework that could perform fast execution on edge devices. AC/DC [44] co-trains the sparse and dense models to return both accurate sparse and dense models. [23] dynamically activates top-K parameters during forward-pass while keeping a larger number of parameters updated in backward-pass to get rid of dense calculation of gradient. Top-KAST [23] preserves constant sparsity throughout training in both the forward and backward passes. Built upon Top-KAST, Powerpropagation [47] leaves the low-magnitude parameters largely unaffected by learning, achieving strong results. CHEX [19] applied dynamic prune and regrow channels strategies to avoid pruning important channels prematurely. Very recently, SLaK [32] leverages dynamic sparse training to successfully train intrinsically sparse $51\times51$ kernels, which performs on par with or better than advanced Transformers. A concurrent work [21] discovers that a tiny fraction of channels (up to 4.3%) of RigL become totally sparse after training.

To enable acceleration of sparse training in practice, [34] build a truly sparse framework based on SciPy sparse matrices [55] that enables efficient sparse evolutionary training [39] in CPU. [6] fulfill group-wise DST on Graphcore IPU [24] and demonstrate its efficacy on pre-training BERT. Moreover, some previous work develops sparse kernels [12; 9] to directly support unstructured sparsity in GPU. DeepSparse [27] deploys large-scale BERT-level and YOLO-level sparse models on CPU.

# 6 Conclusions

In this paper, we have presented **Chase**, a new sparse training approach that seamlessly translates the promise of unstructured sparsity into channel-level sparsity, while performing on par or even often better than state-of-the-art DST approaches. Extensive experiments across various network architectures including VGG-19 and ResNet-50 on CIFAR-10/100 and ImageNet demonstrated Chase can achieve better performance with $1.2\times \sim 1.7\times$ real inference speedup on common GPU devices while performing on par or even better than unstructured SoTA. The results in this paper strongly challenge the common belief that sparse training typically suffers from limited acceleration support in common hardware, opening doors for future work to build more efficient sparse neural networks.

# 7 Acknowledgement

S. Liu and Z. Wang are in part supported by the NSF AI Institute for Foundations of Machine Learning (IFML). Part of this work used the Dutch national e-infrastructure with the support of the SURF Cooperative using grant no. NWO2021.060, EINF-2694 and EINF-2943/L1. It is also supported by the NSF CCF-2312616. Any opinions, findings, and conclusions or recommendations expressed in this material are those of the authors and do not necessarily reflect the views of NSF.

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

# A Remaining Experimental Analysis

## A.1 Effect of the Initial Sparsity

Chase starts from a subnetwork with unstructured sparsity to produce channel-level sparsity during one end-to-end training process. Here we fix the target channel-level sparsity $S_c$ as 0.4 and study how the initialized unstructured sparsity impacts the model performance. The results are reported in Figure 5. It could be seen that, in general, initialization with more parameters leads to better performance. Counter-intuitively, the best accuracy is achieved using 0.6 unstructured sparsity, which is 0.26% higher than initialized as a dense model.

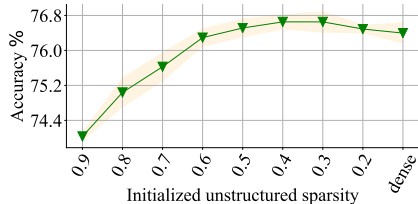

Figure 5: Performance under different initial unstructured sparsity on ResNet-18, CIFAR-100.

## A.2 Effect of the Minimum Channel Ratio Limitation $\beta$

We alter the ratio of the minimum channel ratio $\beta$ to 0.2, 0.5, 0.8 and show the performance in Table 6. The channel pruning frequency $\Delta T$ is fixed at 4000 iterations. Apparently, large ratio $\beta = 0.8$ archives the worst performance while $\beta = 0.2$ outperforms other settings in 4 out of 6 cases. In view of the fact that smaller $\beta$ provide more channel exploration space.

Table 6: Test accuracy (%) on CIFAR-100 of Chase at 50% channel-wise sparsity using different minimum layer limitation factor $\beta$. The best results are marked in bold.

| Minimum | Sparity | | |
|---|---|---|---|
| **layer ratio** | 90% | 95% | 98% |
| VGG-19 | | | |
| 0.20 | **73.45±0.27** | **72.98±0.32** | 71.69±0.21 |
| 0.50 | 73.16±0.06 | 72.39±0.17 | **71.74±0.06** |
| 0.80 | 72.26±0.24 | 72.20±0.27 | 71.28±0.26' |
| ResNet-50 | | | |
| 0.20 | 77.47±0.40 | **77.43±0.36** | **76.68±0.27** |
| 0.50 | **77.54±0.40** | 77.31±0.34 | 76.64±0.26 |
| 0.80 | 76.76±0.66 | 77.26±0.73 | 76.67±0.29 |

## A.3 Ablation of Gradual Amenable Channel Pruning

Here, we perform a more university ablation study of Gradual Amenable Channel Pruning (GACP) on CIFAR10/100, ResNet-50 and VGG-19, with RigL and SET. The results are reported in Table 7. Surprisingly, GACP brings performance increases in most cases. To be specific, we found that GACP could boost the performance of RigL in 9 out of 12 cases and output SET in 10 out of 12 cases with just 50% remaining channels.

## A.4 Effect of the Channel Pruning Frequency

In this Appendix, we study how the channel pruning frequency $\Delta T$ affects Chase's performance. For all experiments, we fixed the ending time $\tau_{stop}$ for gradual amenable channel pruning as 130 epochs, the total training epochs $\tau_{total}$ as 160 epochs and the minimum channel ratio factor as $\beta$ as 0.5, while altering $\Delta T$ to 1000, 4000, 8000, and 16000 iterations. We report the results in Table 8. Overall, the largest $\Delta T$ 16000 leads to worse performance. This observation is as expected, as we aim to achieve the same channel sparsity and larger $\Delta T$ results in more removed channels in each

Table 7: Ablation of gradual amenable channel pruning (GACP). The best results are marked in bold.

| | Dataset | CIFAR-10 | | | CIFAR-100 | | |
|---|---|---|---|---|---|---|---|
| | Sparsity | 90% | 95% | 98% | 90% | 95% | 98% |
| **VGG-19** | SET [39] | 93.61±0.13 | 93.09±0.25 | 91.81±0.04 | 72.58±0.12 | 71.48±0.12 | 69.04±0.15 |
| | SET+GACP ($S_c = 0.5$) | **93.78±0.16** | **93.56±0.05** | **92.66±0.10** | **72.93±0.18** | **71.89±0.20** | **70.03±0.13** |
| | RigL [10] | 93.60±0.09 | 93.05±0.06 | 91.95±0.15 | 72.92±0.31 | **71.85±0.53** | 69.57±0.24 |
| | RigL+GACP ($S_c = 0.5$) | **93.91±0.14** | **93.70±0.06** | **92.80±0.06** | **72.97±0.09** | 71.80±0.17 | **69.83±0.20** |
| **ResNet-50** | SET [39] | **94.65±0.01** | 94.05±0.06 | 92.98±0.18 | 76.14±0.54 | **75.90±0.19** | 73.21±0.06 |
| | SET+GACP ($S_c = 0.5$) | 94.52±0.17 | **94.51±0.19** | **93.23±0.13** | **76.75±0.47** | 75.67±0.64 | **73.86±0.12** |
| | RigL [10] | 94.42±0.17 | 94.22±0.23 | 93.20±0.08 | **77.18±0.42** | **76.50±0.26** | 74.84±0.13 |
| | RigL+GACP ($S_c = 0.5$) | **94.45±0.10** | **94.41±0.13** | **93.50±0.16** | 76.86±0.56 | 76.33±0.63 | **74.92±0.27** |

punning operation. Consequently, larger performance degradation will be introduced during each pruning which could degrade the training stability.

Table 8: Test accuracy (%) on CIFAR-100 of Chase at 50% channel-wise sparsity using different channel pruning frequency $\Delta T$. The best results are marked in bold.

| $\Delta T$ | Sparsity | | |
|---|---|---|---|
| (Iterations) | 90% | 95% | 98% |
| | VGG-19 | | |
| 1000 | 73.07±0.26 | 72.72±0.1 | 71.69±0.35 |
| 4000 | 73.16±0.06 | 72.39±0.17 | 71.74±0.06 |
| 8000 | **73.15±0.23** | **72.81±0.07** | **71.94±0.13** |
| 16000 | 72.88±0.24 | 72.66±0.04 | 71.79±0.43 |
| | ResNet-50 | | |
| 1000 | 77.52±0.30 | 77.48±0.62 | **77.03±0.29** |
| 4000 | **77.54±0.40** | 77.31±0.34 | 76.64±0.26 |
| 8000 | 77.46±0.47 | **77.49±0.47** | 76.74±0.27 |
| 16000 | 77.02±0.25 | 76.96±0.37 | 76.77±0.35 |

# B Implementation Details of Chase

In this appendix, we report the implementation details for Chase, including total training time ($\tau_{total}$), exploration stop time ($\tau_{stop}$), gradual channel pruning frequency ($\Delta T$), parameter update frequency $\Delta T_p$, minimum layer limitation factor ($\beta$), learning rate (LR), batch size (BS), learning rate drop (LR Drop), weight decay (WD), SGD momentum (Momentum), sparse initialization (Sparse Init), target sparsity ($s_p$), target channel-wise sparsity ($S_c$), etc.

Table 9: Implementation hyperparameters of Chase in Table 3, on CIFAR-10/100.

| Model | $\tau_{total}$ (epochs) | $\tau_{stop}$ (epochs) | $\Delta T$ (iterations) | $\Delta T_p$ (iterations) | $\beta$ | BS | LR | LR Drop, Epochs | Optimizer | Momentum | WD | Sparse Init |
|---|---|---|---|---|---|---|---|---|---|---|---|---|
| VGG-19 | 160 | 130 | 8000 | 1000 | 0.2 | 128 | 0.1 | 10x, [80, 120] | SGD | 0.9 | 5e-4 | ERK |
| ResNet-50 | 160 | 130 | 1000 | 1000 | 0.5 | 128 | 0.1 | 10x, [80, 120] | SGD | 0.9 | 5e-4 | ERK |

Table 10: Implementation hyperparameters of Chase in Table 4, on ImageNet.

| Model | $\tau_{total}$ (epochs) | $\tau_{stop}$ (epochs) | $\Delta T$ (iterations) | $\Delta T_p$ (iterations) | $\beta$ | BS | LR | LR Drop | Optimizer | Momentum | WD | Sparse Init |
|---|---|---|---|---|---|---|---|---|---|---|---|---|
| ResNet-50 | 100 | 80 | 1000 | 1000 | 0.2 | 512 | 0.512 | Cosine Decay | SGD | 0.9 | 1e-4 | ERK |
| ResNet-50 | 150 | 120 | 1000 | 1500 | 0.2 | 512 | 0.512 | Cosine Decay | SGD | 0.9 | 1e-4 | ERK |

Table 11: Implementation hyperparameters of Chase in Table 13, on ImageNet.

| Name | $s_p$ | $S_c$ | Model | $\tau_{total}$ (epochs) | $\tau_{stop}$ (epochs) | $\Delta T$ (iterations) | $\Delta T_p$ (iterations) | $\beta$ | BS | LR | LR Drop | Optimizer | Momentum | WD | Sparse Init |
|---|---|---|---|---|---|---|---|---|---|---|---|---|---|---|---|
| Chase-1 | 80% | 40% | ResNet-50 | 250 | 170 | 1000 | 2500 | 0.2 | 512 | 0.512 | Cosine Decay | SGD | 0.9 | 1e-4 | ERK |
| Chase-2 | 90% | 40% | ResNet-50 | 250 | 170 | 1000 | 2500 | 0.2 | 512 | 0.512 | Cosine Decay | SGD | 0.9 | 1e-4 | ERK |

## C Real Inference Speedups

We report the real inference latency and throughput of Chase on various sparity in Table 12.

Table 12: Real inference latency and throughput of Chase on the ResNet-50/ImageNet benchmark. The best results are marked in bold.

| Method | $s_p$ | $S_c$ | Accuracy (%) (↑) | Throughput (↑) | Latency (ms) (↓) |
|---|---|---|---|---|---|
| Chase | 0.9 | 0.2 | **74.40** | 69.30 | 30.89 |
| Chase | 0.9 | 0.3 | 74.35 | 70.07 | 30.39 |
| Chase | 0.9 | 0.4 | 74.03 | 71.54 | 30.13 |
| Chase | 0.9 | 0.5 | 73.39 | 78.19 | 27.55 |
| Chase | 0.9 | 0.6 | 72.85 | 82.98 | 26.09 |
| Chase | 0.9 | 0.7 | 71.98 | **86.26** | **25.10** |
| Chase (prune skip) | 0.9 | 0.2 | **74.15** | 72.57 | 29.38 |
| Chase (prune skip) | 0.9 | 0.3 | 74.06 | 78.30 | 27.33 |
| Chase (prune skip) | 0.9 | 0.4 | 73.24 | 87.38 | 24.53 |
| Chase (prune skip) | 0.9 | 0.5 | 72.71 | 99.97 | 21.55 |
| Chase (prune skip) | 0.9 | 0.6 | 71.62 | 107.99 | 20.03 |
| Chase (prune skip) | 0.9 | 0.7 | 67.15 | **123.30** | **17.56** |
| Chase | 0.8 | 0.2 | **75.82** | 66.39 | 32.05 |
| Chase | 0.8 | 0.3 | 75.62 | 69.30 | 30.81 |
| Chase | 0.8 | 0.4 | 75.27 | 73.32 | 29.22 |
| Chase | 0.8 | 0.5 | 74.76 | 76.68 | 28.05 |
| Chase | 0.8 | 0.6 | 73.77 | 80.51 | 26.81 |
| Chase | 0.8 | 0.7 | 72.88 | **86.12** | **25.15** |
| Chase (prune skip) | 0.8 | 0.2 | **75.27** | 72.47 | 29.38 |
| Chase (prune skip) | 0.8 | 0.3 | 74.96 | 78.60 | 27.20 |
| Chase (prune skip) | 0.8 | 0.4 | 74.58 | 87.91 | 24.38 |
| Chase (prune skip) | 0.8 | 0.5 | 73.53 | 98.43 | 21.86 |
| Chase (prune skip) | 0.8 | 0.6 | 71.70 | 104.82 | 20.58 |
| Chase (prune skip) | 0.8 | 0.7 | 67.53 | **123.46** | **17.57** |

## D Comparisons with SOTA Channel Pruning Methods

We further compare Chase with various state-of-the-art channel pruning approaches in Table 13. It is encouraging to see that Chase performs on par with state-of-the-art SOTA channel pruning approaches, such as Group Fisher [30], CafeNet-R [51], and CHIP [52], without the need for the costly dense pretraining step. The implementation details are reported in Table 13.

Table 13: Comparison with state-of-the-art channel pruning methods on popular benchmark: ResNet-50 on ImageNet.

| Methods | FLOPs | Top-1 | Epochs |
|---|---|---|---|
| GBN [57] | 2.4G | 76.2% | 350 |
| LEGR [4] | 2.4G | 75.7% | - |
| FPGM [16] | 2.4G | 75.6% | 200 |
| TAS [7] | 2.3G | 76.2% | 240 |
| Hrank [29] | 2.3G | 75.0% | 570 |
| SCOP [54] | 2.2G | 76.0% | 230 |
| CHIP [52] | 2.2G | 76.3% | - |
| Group Fisher [30] | 2.0G | 76.4% | - |
| AutoSlim [58] | 2.0G | 75.6% | - |
| Uniform | 2.0G | 75.1% | 300 |
| Random | 2.0G | 74.6% | 300 |
| CafeNet-R [51] | 2.0G | 76.5% | 300 |
| **Chase-1** | 1.5G | 76.6% | 250 |
| Uniform | 1.0G | 73.1% | 300 |
| Random | 1.0G | 72.2% | 300 |
| Group Fisher [30] | 1.0G | 73.9% | - |
| CafeNet-R [51] | 1.0G | 74.9% | 300 |
| CafeNet-E [51] | 1.0G | 75.3% | 300 |
| **Chase-2** | 0.9G | 75.7% | 250 |

# E  Existence of Sparse Amenable Channel in Various Settings

To demonstrate the broad prevalence of sparse amenable channels across diverse architectures and datasets, we have evaluated their ratio in multiple scenarios, including ResNet-32/VGG-16 on CIFAR-100, ResNet-50/ViT-small on ImageNet, an MLP model on CIFAR-10.

The MLP model consists of two hidden layers, each with 512 neurons. For the ViT small model, we focus our attention on the neurons within the MLP layers that exhibit suitability for pruning. In all cases, the sparse amenable channels are identified by *Unmasked Mean Magnitude* (UMM), with the threshold, $v$, set to 20%.

The results, shown in the corresponding tables, underline the consistent presence of sparse amenable channels across various architectures and datasets, reinforcing the argument that the phenomenon is both significant and widespread.

Table 14: Sparse amenable channel portion during training on various settings

| Settings | Layer | 10 Epoch | 20 Epoch | 40 Epoch | 100 Epoch |
|---|---|---|---|---|---|
| ResNet-32/CIFAR-100 | blocks.3.conv1 | 0.23 | 0.38 | 0.38 | 0.63 |
| | blocks.6.conv1 | 0.16 | 0.19 | 0.28 | 0.32 |
| VGG-16/CIFAR-100 | features.0 | 0.11 | 0.22 | 0.30 | 0.39 |
| | features.7 | 0.19 | 0.20 | 0.17 | 0.65 |

| Settings | Layer | 5 Epoch | 10 Epoch | 20 Epoch | 50 Epoch |
|---|---|---|---|---|---|
| ResNet-50/ImageNet | layer3.1.conv2 | 0.14 | 0.18 | 0.21 | 0.33 |
| | layer4.1.conv1 | 0.44 | 0.49 | 0.49 | 0.56 |
| ViT-Small/ImageNet | blocks.0.mlp.fc1 | 0.11 | 0.14 | 0.15 | 0.31 |
| | blocks.8.mlp.fc1 | 0.20 | 0.22 | 0.21 | 0.33 |

| Settings | Layer | 5 Epoch | 10 Epoch | 20 Epoch | 50 Epoch |
|---|---|---|---|---|---|
| MLP Model/Cifar10 | fc1 | 0.21 | 0.21 | 0.30 | 0.37 |
| | fc2 | 0.36 | 0.42 | 0.49 | 0.63 |

# F    Addressing the Memory Limitation of Global Parameter Exploration

During global parameter exploration, directly loading all the parameters for gradients/magnitude sorting is memory-consuming. Inspired by [40], we layer-wisely select parameters with the largest gradients for growth and the lowest magnitude for pruning by an adaptive global threshold $H$, until reaching the target sparsity. $H$ is determined by a set point negative feedback loop to maintain an approximate parameter amount during each reallocation step, as reported below.

---

**Algorithm 2:** Overview of Global Parameter Exploration

---

**Input:**  Network with sparse weight $\boldsymbol{\theta}_s$, target sparsity $s_p$, current sparsity $s_t$, prune magnitude threshold $H_p$, grow gradient threshold $H_g$, threshold incremental value $H_i$, sparsity tolerance $s_\delta$

**Output:** A sparse model $\boldsymbol{\theta}_s$ satisfying the target sparsity $s_p$.

Initialize a grow threshold $H_g$        ▷ *Begin global parameter growing*

**while** ***not*** $s_p + s_\delta > s_t > s_p - s_\delta$ **do**

    **for** *each sparse tensor $\boldsymbol{\theta}_s^l$ of layer $l$* **do**

        $(\boldsymbol{\theta}_s^l, g_l) \leftarrow \texttt{grow\_by\_threshold}(\boldsymbol{\theta}_s^l, H_g)$     ▷ $g_l$ *is the number of pruned weights in layer $l$*

    $G \leftarrow \sum_i g_i$ , $s_t \leftarrow \texttt{calculate\_current\_sparsity}(G)$ ▷ $G$ *is total number of grown weights in all layers*

    **if** $s_t < s_p - s_\delta$ **then**

        $H_g \leftarrow (H_g + H_i)$

    **if** $s_t > s_p + s_\delta$ **then**

        $H_g \leftarrow (H_g - H_i)$            ▷ *Update the grow threshold*

Initialize a prune threshold $H_p$        ▷ *Begin global parameter pruning*

**while** ***not*** $s_p + s_\delta > s_t > s_p - s_\delta$ **do**

    **for** *each sparse tensor $\boldsymbol{\theta}_s^l$ of layer $l$* **do**

        $(\boldsymbol{\theta}_s^l, p_l) \leftarrow \texttt{prune\_by\_threshold}(\boldsymbol{\theta}_s^l, H_p)$     ▷ $p_l$ *is the number of pruned weights in layer $l$*

    $P \leftarrow \sum_i p_i$ , $s_t \leftarrow \texttt{calculate\_current\_sparsity}(P)$     ▷ $P$ *is the total number of pruned weights in all layers*

    **if** $s_t < s_p - s_\delta$ **then**

        $H_p \leftarrow (H_p - H_i)$

    **if** $s_t > s_p + s_\delta$ **then**

        $H_p \leftarrow (H_p + H_i)$            ▷ *Update the prune threshold*

---

# G   Learned Layerwise Sparsity

Table 15 summarize the final learned layer-wise sparsity on ResNet-50 under 80% sparsity and 40% channel-wise sparsity. The model is obtained by training 100 epochs on ImageNet-1K. The parameter-wise sparsity represents the sparsity budgets for all the CNN layers without the last fully-connected layer. The channel-wise sparsity denotes the sparsity of all the CNN internal layers (the first two convolution layers in the bottleneck blocks).

Table 15: ResNet-50 learnt budgets using Chase at 80% and channel-wise sparsity at 40%.

| ResNet-50 | Fully Dense Params | Fully Dense Weights Dimension | Weights Dimension after Chase | Parameter-wise Sparsity (%) | Channel-wise Sparsity (%) |
|---|---|---|---|---|---|
| Backbone | 23454912 | - | - | 80 | 40 |
| Layer 1 - conv1 | 9408 | $64 \times 3 \times 7 \times 7$ | $64 \times 3 \times 7 \times 7$ | 5.11 | 0.00 |
| Layer 2 - layer1.0.conv1 | 4096 | $64 \times 64 \times 1 \times 1$ | $64 \times 64 \times 1 \times 1$ | 8.45 | 0.00 |
| Layer 3 - layer1.0.conv2 | 36864 | $64 \times 64 \times 3 \times 3$ | $64 \times 64 \times 3 \times 3$ | 44.61 | 0.00 |
| Layer 4 - layer1.0.conv3 | 16384 | $256 \times 64 \times 1 \times 1$ | $256 \times 64 \times 1 \times 1$ | 51.16 | 0.00 |
| Layer 5 - layer1.0.downsample.0 | 16384 | $256 \times 64 \times 1 \times 1$ | $256 \times 64 \times 1 \times 1$ | 65.58 | 0.00 |
| Layer 6 - layer1.1.conv1 | 16384 | $64 \times 256 \times 1 \times 1$ | $63 \times 256 \times 1 \times 1$ | 53.13 | 1.56 |
| Layer 7 - layer1.1.conv2 | 36864 | $64 \times 64 \times 3 \times 3$ | $62 \times 63 \times 3 \times 3$ | 42.51 | 3.13 |
| Layer 8 - layer1.1.conv3 | 16384 | $256 \times 64 \times 1 \times 1$ | $256 \times 62 \times 1 \times 1$ | 36.48 | 0.00 |
| Layer 9 - layer1.2.conv1 | 16384 | $64 \times 256 \times 1 \times 1$ | $63 \times 256 \times 1 \times 1$ | 48.64 | 1.56 |
| Layer 10 - layer1.2.conv2 | 36864 | $64 \times 64 \times 3 \times 3$ | $64 \times 63 \times 3 \times 3$ | 52.23 | 0.00 |
| Layer 11 - layer1.2.conv3 | 16384 | $256 \times 64 \times 1 \times 1$ | $256 \times 64 \times 1 \times 1$ | 48.68 | 0.00 |
| Layer 12 - layer2.0.conv1 | 32768 | $128 \times 256 \times 1 \times 1$ | $128 \times 256 \times 1 \times 1$ | 32.51 | 0.00 |
| Layer 13 - layer2.0.conv2 | 147456 | $128 \times 128 \times 3 \times 3$ | $127 \times 128 \times 3 \times 3$ | 71.73 | 0.78 |
| Layer 14 - layer2.0.conv3 | 65536 | $512 \times 128 \times 1 \times 1$ | $512 \times 127 \times 1 \times 1$ | 54.16 | 0.00 |
| Layer 15 - layer2.0.downsample.0 | 32768 | $512 \times 256 \times 1 \times 1$ | $512 \times 256 \times 1 \times 1$ | 86.72 | 0.00 |
| Layer 16 - layer2.1.conv1 | 65536 | $128 \times 512 \times 1 \times 1$ | $111 \times 512 \times 1 \times 1$ | 79.24 | 13.28 |
| Layer 17 - layer2.1.conv2 | 147456 | $128 \times 128 \times 3 \times 3$ | $117 \times 111 \times 3 \times 3$ | 73.93 | 8.59 |
| Layer 18 - layer2.1.conv3 | 65536 | $512 \times 128 \times 1 \times 1$ | $512 \times 117 \times 1 \times 1$ | 69.68 | 0.00 |
| Layer 19 - layer2.2.conv1 | 65536 | $128 \times 512 \times 1 \times 1$ | $106 \times 512 \times 1 \times 1$ | 79.15 | 17.19 |
| Layer 20 - layer2.2.conv2 | 147456 | $128 \times 128 \times 3 \times 3$ | $108 \times 106 \times 3 \times 3$ | 74.98 | 15.62 |
| Layer 21 - layer2.2.conv3 | 65536 | $512 \times 128 \times 1 \times 1$ | $512 \times 108 \times 1 \times 1$ | 74.50 | 0.00 |
| Layer 22 - layer2.3.conv1 | 65536 | $128 \times 512 \times 1 \times 1$ | $122 \times 512 \times 1 \times 1$ | 78.87 | 4.69 |
| Layer 23 - layer2.3.conv2 | 147456 | $128 \times 128 \times 3 \times 3$ | $96 \times 122 \times 3 \times 3$ | 78.88 | 25.00 |
| Layer 24 - layer2.3.conv3 | 65536 | $512 \times 128 \times 1 \times 1$ | $512 \times 96 \times 1 \times 1$ | 71.11 | 0.00 |
| Layer 25 - layer3.0.conv1 | 131072 | $256 \times 512 \times 1 \times 1$ | $255 \times 512 \times 1 \times 1$ | 57.36 | 0.39 |
| Layer 26 - layer3.0.conv2 | 589824 | $256 \times 256 \times 3 \times 3$ | $241 \times 255 \times 3 \times 3$ | 85.19 | 5.86 |
| Layer 27 - layer3.0.conv3 | 262144 | $1024 \times 256 \times 1 \times 1$ | $1024 \times 241 \times 1 \times 1$ | 65.11 | 0.00 |
| Layer 28 - layer3.0.downsample.0 | 524288 | $1024 \times 512 \times 1 \times 1$ | $1024 \times 512 \times 1 \times 1$ | 97.36 | 0.00 |
| Layer 29 - layer3.1.conv1 | 262144 | $256 \times 1024 \times 1 \times 1$ | $151 \times 1024 \times 1 \times 1$ | 85.45 | 41.80 |
| Layer 30 - layer3.1.conv2 | 589824 | $256 \times 256 \times 3 \times 3$ | $143 \times 151 \times 3 \times 3$ | 75.48 | 44.14 |
| Layer 31 - layer3.1.conv3 | 262144 | $1024 \times 256 \times 1 \times 1$ | $1024 \times 143 \times 1 \times 1$ | 76.19 | 0.00 |
| Layer 32 - layer3.2.conv1 | 262144 | $256 \times 1024 \times 1 \times 1$ | $151 \times 1024 \times 1 \times 1$ | 83.22 | 41.80 |
| Layer 33 - layer3.2.conv2 | 589824 | $256 \times 256 \times 3 \times 3$ | $108 \times 151 \times 3 \times 3$ | 77.29 | 57.81 |
| Layer 34 - layer3.2.conv3 | 262144 | $1024 \times 256 \times 1 \times 1$ | $1024 \times 108 \times 1 \times 1$ | 76.97 | 0.00 |
| Layer 35 - layer3.3.conv1 | 262144 | $256 \times 1024 \times 1 \times 1$ | $91 \times 1024 \times 1 \times 1$ | 84.94 | 64.84 |
| Layer 36 - layer3.3.conv2 | 589824 | $256 \times 256 \times 3 \times 3$ | $74 \times 91 \times 3 \times 3$ | 63.20 | 71.88 |
| Layer 37 - layer3.3.conv3 | 262144 | $1024 \times 256 \times 1 \times 1$ | $1024 \times 74 \times 1 \times 1$ | 78.42 | 0.00 |
| Layer 38 - layer3.4.conv1 | 262144 | $256 \times 1024 \times 1 \times 1$ | $76 \times 1024 \times 1 \times 1$ | 84.05 | 70.31 |
| Layer 39 - layer3.4.conv2 | 589824 | $256 \times 256 \times 3 \times 3$ | $41 \times 76 \times 3 \times 3$ | 56.02 | 83.98 |
| Layer 40 - layer3.4.conv3 | 262144 | $1024 \times 256 \times 1 \times 1$ | $1024 \times 41 \times 1 \times 1$ | 73.70 | 0.00 |
| Layer 41 - layer3.5.conv1 | 262144 | $256 \times 1024 \times 1 \times 1$ | $105 \times 1024 \times 1 \times 1$ | 83.33 | 58.59 |
| Layer 42 - layer3.5.conv2 | 589824 | $256 \times 256 \times 3 \times 3$ | $44 \times 105 \times 3 \times 3$ | 63.34 | 82.81 |
| Layer 43 - layer3.5.conv3 | 262144 | $1024 \times 256 \times 1 \times 1$ | $1024 \times 44 \times 1 \times 1$ | 66.12 | 0.00 |
| Layer 44 - layer4.0.conv1 | 524288 | $512 \times 1024 \times 1 \times 1$ | $499 \times 1024 \times 1 \times 1$ | 74.52 | 2.54 |
| Layer 45 - layer4.0.conv2 | 2359296 | $512 \times 512 \times 3 \times 3$ | $241 \times 499 \times 3 \times 3$ | 88.06 | 52.93 |
| Layer 46 - layer4.0.conv3 | 1048576 | $2048 \times 512 \times 1 \times 1$ | $2048 \times 241 \times 1 \times 1$ | 65.89 | 0.00 |
| Layer 47 - layer4.0.downsample.0 | 2097152 | $2048 \times 1024 \times 1 \times 1$ | $2048 \times 1024 \times 1 \times 1$ | 99.55 | 0.00 |
| Layer 48 - layer4.1.conv1 | 1048576 | $512 \times 2048 \times 1 \times 1$ | $146 \times 2048 \times 1 \times 1$ | 84.56 | 71.09 |
| Layer 49 - layer4.1.conv2 | 2359296 | $512 \times 512 \times 3 \times 3$ | $89 \times 146 \times 3 \times 3$ | 62.39 | 82.23 |
| Layer 50 - layer4.1.conv3 | 1048576 | $2048 \times 512 \times 1 \times 1$ | $2048 \times 89 \times 1 \times 1$ | 72.09 | 0.00 |
| Layer 51 - layer4.2.conv1 | 1048576 | $512 \times 2048 \times 1 \times 1$ | $489 \times 2048 \times 1 \times 1$ | 82.31 | 4.49 |
| Layer 52 - layer4.2.conv2 | 2359296 | $512 \times 512 \times 3 \times 3$ | $292 \times 489 \times 3 \times 3$ | 87.01 | 42.97 |
| Layer 53 - layer4.2.conv3 | 1048576 | $2048 \times 512 \times 1 \times 1$ | $2048 \times 292 \times 1 \times 1$ | 62.42 | 0.00 |
| Layer 54 - fc | 2048000 | $1000 \times 2048$ | $1000 \times 2048$ | 61.82 | 0.00 |

