# OpenReview forum: "Dynamic Sparsity Is Channel-Level Sparsity Learner"
_NeurIPS.cc/2023/Conference — NeurIPS 2023 poster_

### Official Review · Reviewer_DQ24 · 2023-06-25

**Soundness:** 3 good
**Presentation:** 4 excellent
**Contribution:** 4 excellent
**Rating:** 7
**Confidence:** 5

**Summary:**

This paper investigates practical deployment and speedup of Dynamic Sparse Training methods on general hardwares. The author's motivation stems from observations that many channels naturally exhibit high sparsity rates. Based on this finding, the authors propose a sparse-inducing method to gradually remove entire channels during sparse training. This ultimately produces a network with certain structured sparse characteristics, enabling favorable acceleration. The authors validate the effectiveness of their proposed method through experiments on extensive datasets.

**Strengths:**

1. The motivation is highly reasonable, and the proposed method has great application prospects, as it overcomes the bottleneck of unstructured sparsity being difficult to deploy on off-the-shelf hardware.
2. The observation on the sparsity distribution across channels of DST is novel and provides inspiration for the community.
3. The proposed method is easy to implement and even surpasses traditional non-structured pruning methods while achieving practical GPU acceleration.


**Weaknesses:**

1. Given the author's focus on practical acceleration in the pruning field, it would be interesting to compare this method with structured pruning methods [1,2] even after obtaining accelerated and deployable networks.

[1] Pruning filters for efficient ConvNets. In ICLR, 2017.

[2] Channel Pruning via Automatic Structure Search. In IJCAI, 2020.

**Questions:**

Regarding weight regrowing, I understand that the author selects weights with the highest magnitude globally for restoration. How about restoring weights with the highest UMM in each channel? Would this allow the network to gradually learn to become completely structured, i.e., some channels are automatically removed, while important channels are all preserved?

---

> ### Author Rebuttal · Authors · 2023-08-10
>
>
> We sincerely appreciate your encouraging feedback. It's heartening to hear that you find the motivation compelling and recognize the potential application of our method. We are especially grateful for your acknowledgment of the novelty in our observation on sparsity distribution and its potential to inspire the community.  We provide the following response to address your concerns point-by-point:
>
> **Comment 1:  Given the author's focus on practical acceleration in the pruning field, it would be interesting to compare this method with structured pruning methods [1,2] even after obtaining accelerated and deployable networks.**
>
>
> - Thanks for bringing these two interesting works, we compare Chase with these two baselines in the following table. It could be seen that Chase consistently outperforms these two baselines while achieving much fewer inference FLOPs, even though being a sparse-to-sparse training method.
> | Model     | Approach                     | Top-1 Acc (\%) | Inference FLOPs ×(10e9) |
> |:-----------|------------------------------|----------------|-------------------------|
> | ResNet-50 | Chase ($S_c=0.4$)            | **74.34**      | 1.80                    |
> |  ResNet-50   | ABC Pruner [Lin et al.]  | 73.52          | 1.79                    |
> | ResNet-34 | Chase ($S_c=0.4$)            | **72.61**      | 1.70                    |
> |  ResNet-34  | ABCPruner [Lin et al.]       | 70.98          | 2.17                    |
> | ResNet-34  | Efficient Convnets [Li et al.] | 72.56        | 3.08                    |
>
> [1] Li H, Kadav A, Durdanovic I, et al. Pruning filters for efficient convnets.In ICLR, 2017.
>
> [2] Lin M, Ji R, Zhang Y, et al. Channel pruning via automatic structure search .In IJCAI, 2020.
>
>
> **Comment 2: Regarding weight regrowing, I understand that the author selects weights with the highest magnitude globally for restoration. How about restoring weights with the highest UMM in each channel? Would this allow the network to gradually learn to become completely structured, i.e., some channels are automatically removed, while important channels are all preserved?**
>
>
> - We appreciate your insightful suggestion. Targeting channels with higher UMM values could indeed be a promising strategy for restoring weights.
>
> - After seeing your great suggestion, we make a deep dive into how the new weights are restored during training and surprisingly found that the majority of new weights are actually restored from the channels with high UMM scores.  To demonstrate this, we conducted an experiment with the following settings: we utilized Global gradients as the criteria to train a 90\% sparse ResNet-50, emulating the Chase methodology. We first rank channels based on their UMM scores and split them into two categories: Large UMM channels: top-50\% channels with the highest UMM scores; Small UMM channels: the rest 50\% channels with the lowest UMM scores. Subsequently, we compute the proportion of newly restored weights within these two categories at different training epochs respectively.
>
> - Our results, detailed in the following table, reveal that Chase actually regrows the majority of the regrown weights (up to 91\%) in channels with higher UMM scores. This result highly aligns with your idea of restoring weights with the highest UMM scores. We sincerely appreciate your knowledgeable contribution and will leave this as promising future work.
> | Epoch | Weights restored in small UMM channels (\%) | Weights restored in large UMM channels (\%) |
> |-------|---------------------------------------------|---------------------------------------------|
> | 20    | 39.06                                       | 60.94                                       |
> | 40    | 38.31                                       | 61.69                                       |
> | 60    | 40.76                                       | 59.24                                       |
> | 80    | 31.60                                       | 68.40                                       |
> | 100   | 25.77                                       | 74.23                                       |
> | 120   | 08.28                                        | 91.72                                       |
> | 140   | 14.27                                       | 85.73                                       |
> | 160   | 19.39                                       | 80.61                                       |

---

> > ### Comment · Reviewer_DQ24 · 2023-08-11
> > **Reply**
> >
> > Thanks for the rebuttal. Overall, I am satisfied with the quality of this paper and believe it is worth publishing due to its contribution on deploying unstructured sparsity methods on off-the-shelf hardware.

---

> > > ### Author Response · Authors · 2023-08-11
> > > **Thanks**
> > >
> > > Dear Reviewer DQ24,
> > >
> > > We would like to express our gratitude for your constructive comments and support! We are so glad that you recognize the value of our paper to the community.
> > >
> > > Best,
> > >
> > > Authors

---

### Official Review · Reviewer_G8BR · 2023-07-05

**Soundness:** 3 good
**Presentation:** 3 good
**Contribution:** 3 good
**Rating:** 7
**Confidence:** 5

**Summary:**

This paper proposes a new approach to dynamic sparsity called Channel-aware dynamic sparse (Chase), which gradually translates unstructured dynamic sparsity to GPU-friendly channel-level sparsity during one end-to-end sparse training process.

**Strengths:**

-	The technical novelty of Chase mainly lies in two aspects: on the structured sparsity level, it adopts the gradual sparsification schedule to gradually remove Amenable Channels during training with smallest UMM scores, and on the unstructured sparsity level, it globally redistributes parameters based on their magnitude and gradient, which significantly strengthens the sparse training performance. I am a fan of this main idea.

-	Chase differs from other approaches to dynamic sparsity in that it seamlessly translates unstructured dynamic sparsity to GPU-friendly channel-level sparsity during one end-to-end training pass, without any ad-hoc operations. Therefore, it is designed to be a more efficient and effective approach to dynamic sparsity that can be accelerated by commodity hardware. This is in contrast to other approaches demonstrated on unstructured sparsity which receive limited support in common hardware.

-	One of the main new observations is that sparse amenable channels, which are channels that rapidly become sparser than others at the very early training stage, cause marginal damages to the model performance than their counterparts when pruned. The paper proposes a new metric called Unmasked Mean Magnitude (UMM) that can be used to precisely discover sparse amenable channels during training by monitoring the quantity and quality of weight sparsity. This finding reveals that a more nuanced approach to channel pruning can be more effective in achieving sparsity in neural networks.

-	Chase achieves up to 1.7× inference throughput speedup on common GPU devices without compromising accuracy with ResNet-50 on ImageNet. Even without pruning skip connections, the model is able to provide 31% throughput speedups, while outperforming the SOTA RigL by 0.39%. Additionally, Chase establishes a new SOTA performance bar for unstructured sparse training, outperforming all the presented DST methods by good margins.


**Weaknesses:**

-	Chase appears to be a mixture between “unstructured sparse training” and “structured sparse inference”. So, what is the benefit of Chase over early-stage structured pruning methods, such as Early Bird lottery ticket? How about its comparison to N:M dynamic sparse training, which can be done now from end to end? And how about instead just training a slimmer dense network from scratch? Those important questions are not discussed.

-	Experiment choices seem flawed to me. Both VGG19 and ResNet50 are known to be (ridiculously) over-parameterized for CIFAR-10/100. Comparison results on those testbeds cannot be reliable for sparsity algorithms: for CIFAR perhaps ResNet-32 or Res-20 would have made more sense. The only persuasive setting in this paper is ResNet 50 + ImageNet.

-	Important detail is missing on how “GPU-supported FLOPs” is computed? On what GPU? By channel level or N:M level? Also, how is the “Throughput” computed in Table 5?

-	Chase requires additional memory to store the mask for each channel. This can be a limiting factor for models with a large number of channels, as the memory requirements can become prohibitively large.

-	Additionally, the paper does not provide a detailed analysis of the end-to-end computational cost of Chase compared to other dynamic sparse training methods, which could be an important consideration for practical applications.


**Questions:**

Please see above weakness.

**Limitations:**

The authors shall discuss more limitations of their method

---

> ### Author Rebuttal · Authors · 2023-08-10
>
>
> We greatly appreciate your positive feedback and we are glad that you like our idea. We provide the following response to address your concerns point-by-point.
>
>
> **Comment 1: Chase appears to be a mixture between “unstructured sparse training” and “structured sparse inference”. So, what is the benefit of Chase over early-stage structured pruning methods, such as Early Bird lottery ticket? How about its comparison to N:M dynamic sparse training, which can be done now from end to end? And how about instead just training a slimmer dense network from scratch? Those important questions are not discussed.**
>
> Thank you for providing us with such valuable and constructive comments, which certainly contributed to enhancing the positioning of our paper. We provide our discussion below.
>
> - Chase is essentially a sparse-to-sparse training approach that does not require any pre-training or dense training steps, i.e., starting from unstructured sparse and ending up with structured sparse. In contrast, the Early Bird Lottery Ticket (EB tickets [1]) follows a dense-to-sparse training paradigm. Despite the fact that the winning ticket can be discovered much earlier than the original Lottery Ticket approach, a certain duration of dense training is necessitated—typically ranging from 6\% to 20\% of the full training time [1]. In addition, when initiating from an 80\% unstructured ResNet-50, Chase surpasses EB tickets by a significant margin of 2.01\% (achieving 75.87\% accuracy compared to 73.86\%) under a 30\% channel sparsity, showing stronger results.
> - Chase presents a more hardware-friendly alternative in comparison to the N:M sparsity technique. While the N:M approach holds substantial promise, it remains confined to a select range of hardware, specifically the NVIDIA Ampere Architecture. Furthermore, only the 2:4 pattern is currently amenable to acceleration. Conversely, Chase employs a channel-wise sparse pattern, which seamlessly interfaces with readily available hardware, obviating the need for specialized sparsity-aware accelerators.
>
> - It is commonly observed that a slimmer dense network, often referred to as a ``small dense`` model, typically falls short in performance compared to a larger sparse model trained through RigL [2]. However, Chase consistently performs on par, or even better than RigL with up to 1.7× inference throughput speedup, as exemplified in our submission. Consequently, Chase offers substantial accuracy advantages over a slimmer dense network.
>
>
>
> [1] You, Haoran, et al. "Drawing early-bird tickets: Towards more efficient training of deep networks." arXiv preprint arXiv:1909.11957 (2019).
>
> [2] Evci, Utku, et al. "Rigging the Lottery: Making All Tickets Winners. arXiv e-prints, art." arXiv preprint arXiv:1911.11134 (2019).
>
>
> **Comment 2: For CIFAR perhaps ResNet-32 or Res-20 would have made more sense**
>
> - Thanks for your suggestion.  We want to highlight that we have actually included the results of ResNet-20 on CIFAR-10/100 in our submission at the bottom of Table 3. **We also added extra experiments of ResNet-32 and report all the results in Table 1 of the rebuttal pdf for your convenience.** As we can see, Chase again significantly outperforms RigL at all sparsity levels, demonstrating its effectiveness on smaller architectures. We will highlight the results of ResNet-20 and ResNet-32 in the camera-ready version.
>
>
>
> **Comment 3: Important detail is missing on how “GPU-supported FLOPs” is computed? On what GPU? By channel level or N:M level? Also, how is the “Throughput” computed in Table 5**
>
>
> - Thanks for your asking.  "GPU-supported FLOPs" refers to the Floating Point Operations (FLOPs) that are measured on common GPUs without any specific sparsity-aware accelerators where channels containing unstructured sparsity are usually treated as a dense channel. This approach mirrors the inference of sparse models on common GPUs without any specific sparsity-aware accelerators.
>
> - The throughput, denoted as images/second, indicates the number of images processed by models per second during inference [1, 2]. It was measured on an NVIDIA 2080TI GPU device, using float32 precision within PyTorch.
>
>
> [1] Liu Z, Mao H, Wu C Y, et al. A convnet for the 2020s, CVPR. 2022
>
> [2] Liu Z, Lin Y, Cao Y, et al. Swin transformer: Hierarchical vision transformer using shifted windows, CVPR 2021:
>
> **Comment 4: Chase requires additional memory to store the mask for each channel. This can be a limiting factor for models with a large number of channels, as the memory requirements can become prohibitively large.**
>
> - You are correct that Chase requires storing the mask for each channel. Fortunately, since additional masks are already used by other sparse training approaches such as RigL and SET, we directly leverage those masks to perform channel pruning without using additional memory. Therefore, we would like to clarify that Chase does not need extra memory to store masks compared to existing sparse training approaches.
>
>
>
>
> **Comment 5: Detailed analysis of the end-to-end computational cost of Chase compared to other dynamic sparse training methods**
>
> - Please refer to our global responses.
>
>
> **Comment 6:  The authors shall discuss more limitations of their method**
>
> - We appreciate the reviewer pointing this out. One limitation of our method is its dependence on Chase's gradual channel pruning, which gradually prunes channels to reach the desired channel sparsity. As a result, the peak memory requirement remains equivalent to that of a dense network. Moving forward, we're considering the integration of one-shot channel pruning prior to training, which we anticipate could greatly enhance efficiency.
>
> - Additionally, for the sake of convenience and ease of implementation, we currently employ masks for channel pruning during training. To truly realize training speed-up, more engineering efforts are warranted, which we plan to undertake in future iterations

---

> > ### Comment · Reviewer_G8BR · 2023-08-11
> > **Thanks**
> >
> > Thanks for the rebuttal. My concerns are well addressed and I think the paper is ready for publishing. I increase my score to Accept.

---

> > > ### Author Response · Authors · 2023-08-11
> > > **Thanks for your support**
> > >
> > > Dear Reviewer G8BR,
> > >
> > > Thanks for increasing your score. Your support means a lot to us. We really appreciate it!
> > >
> > > Please let us know if you have additional advice or changes.
> > >
> > > Best,
> > >
> > > Authors

---

### Official Review · Reviewer_VN42 · 2023-07-06

**Soundness:** 3 good
**Presentation:** 3 good
**Contribution:** 2 fair
**Rating:** 6
**Confidence:** 4

**Summary:**

This paper proposes Channel-aware dynamic sparse, which translates unstructured dynamic sparsity to GPU-friendly channel-level sparsity during one end-to-end training process without using any particularly sparsity-aware hardware accelerators. The authors also observe the sparse amendable channels and proposed a new metric - Unmasked Mean Magnitude. Experiments demonstrate that the proposed method achieves 1.2x - 1.7x inference throughput speedups on common GPU devices with less to no accuracy degradation.

**Strengths:**

1. This paper is well-written and gives sufficient experiments.
2. This paper observes the channel-wise sparsity during dynamic sparse training and proposes a framework Chase to translate unstructured dynamic sparsity to channel-level sparsity with less to no accuracy loss.

**Weaknesses:**

1. The proposed method does not seem to have much performance benefit for training compared to unstructured methods. The computation is still unstructured in most iterations.

2. In Figure 3-b, the accuracy of RigL+GACP and RigL+GACP+SM is lower than RigL for ResNet-50 with sparsity 90% on CIFAR-100. While I understand GACP improves efficiency, it is not a good way to show the advantage of GACP as accuracy drops. It will be more convincing if you can show the runtime at the same accuracy or the accuracy with the same runtime.






**Questions:**

The experiments use 50% channel sparsity. Why 50%?

**Limitations:**

I don't find a discussion on the limitations.

---

> ### Author Rebuttal · Authors · 2023-08-10
>
>
> We would like to thank you for the time to review our work and are glad that you find our work is well-written with sufficient experiments. We would like to address all the weaknesses pointed out by you below:
>
> **Comment 1: . The proposed method does not seem to have much performance benefit for training compared to unstructured methods. The computation is still unstructured in most iterations..**
>
> Thank you for providing us with such valuable and constructive comments, which certainly contributed to enhancing the positioning of our paper. We provide our discussion below.
>
>
> - Please allow us to clarify that the channel pruning of Chase starts to perform at the very beginning of training, i.e., the **first epoch** for ResNet-50/ImageNet, thanks to the emergence of channel-wise sparsity implicitly produced by DST. To provide further evidence, we accumulate the reduction of training FLOPs solely caused by channel pruning, without considering unstructured/weight sparsity. With a 40\% channel sparsity, Chase is able to achieve a 1.4$\times$ training speedup when training ResNet-50 on ImageNet for 100 epochs. We expect to receive a higher speedup if we consider the benefits of unstructured sparsity together, which is widely accepted by the  community.
>
>
> **Comment 2: In Figure 3-b, the accuracy of RigL+GACP and RigL+GACP+SM is lower than RigL for ResNet-50 with sparsity 90\% on CIFAR-100. While I understand GACP improves efficiency, it is not a good way to show the advantage of GACP as accuracy drops. It will be more convincing if you can show the runtime at the same accuracy or the accuracy with the same runtime.**
>
>
> - We really appreciate your constructive comment, which indeed helps us demonstrate the advantages of our approach. **As you suggested, we've detailed the throughput-accuracy trade-offs for different methods in Figure 1 of the rebuttal pdf.** High throughput indicates shorter runing time. To draw a more strong conclusion, we added a new ablation study with ResNet-50 on ImageNet. We can see that at the same accuracy, RigL+GACP offers around 30\% throughput gains compared to RigL, while maintaining comparable performance. RigL+GACP+SM is able to push the throughput gains to 37\% and the full version of Chase leads to a 60\% reduction eventually, demonstrating the advantages of our approach.
>
> **Comment 3 : The experiments use 50\% channel sparsity. Why 50\%?**
>
>
> - Thank you for your question. I'd like to clarify that our submission goes beyond a mere selection of a 50\% sparsity level. In fact, we have reported a diverse spectrum of sparsity values, ranging from 20\% to 70\%, across various experiments involving CIFAR and ImageNet, depicted in Figure 2, Figure 5 (Appendix), and Table 8 (Appendix).
>
> - We choose 50\% channel sparsity in Table 4 as we firmly believe that a 50\% sparsity level effectively showcases Chase's capability to eliminate a significant proportion of channels, thereby substantiating its efficiency, which is widely used in channel pruning papers [1,2,3].
>
> [1] He, Yihui, Xiangyu Zhang, and Jian Sun. "Channel pruning for accelerating very deep neural networks." Proceedings of the IEEE international conference on computer vision. 2017.
>
> [2] Luo, Jian-Hao, Jianxin Wu, and Weiyao Lin. "Thinet: A filter level pruning method for deep neural network compression." Proceedings of the IEEE international conference on computer vision. 2017.
>
> [3] Raihan, Md Aamir, and Tor Aamodt. "Sparse weight activation training." Advances in Neural Information Processing Systems 33 (2020): 15625-15638.
>
>
> **Comment 4: I don't find a discussion on the limitations**
>
> - We appreciate the reviewer pointing this out. One limitation of our method is its reliance on Chase's gradual channel pruning, which gradually trims channels until the target channel sparsity is achieved. Consequently, the maximum memory usage is on par with a fully dense network. In future iterations, we are exploring the incorporation of one-shot channel pruning before training, which could significantly improve efficiency.

---

> > ### Comment · Reviewer_VN42 · 2023-08-11
> >
> > Thank you for the rebuttal. It has addressed most of my concerns. I agree with other reviewers that the paper is ready for publication.

---

> > > ### Author Response · Authors · 2023-08-11
> > > **Thanks**
> > >
> > > Dear Reviewer VN42,
> > >
> > > We sincerely thank you for raising your score. We hold your support in high regard! If you possess further insights or advice, your input is warmly welcomed.
> > >
> > > Best regards,
> > > The Authors

---

### Official Review · Reviewer_bsGW · 2023-07-07

**Soundness:** 3 good
**Presentation:** 3 good
**Contribution:** 2 fair
**Rating:** 6
**Confidence:** 2

**Summary:**

The paper introduces a novel approach called Channel-aware dynamic sparse (Chase) for training deep neural networks with high sparsity. Unlike previous techniques that focus on unstructured sparsity with irregular patterns, Chase seamlessly translates the benefits of dynamic sparsity to channel-level sparsity, which is more hardware-friendly. This is achieved through an end-to-end training process without requiring additional ad-hoc operations or specialized hardware accelerators. By identifying and removing sparser channels during training, Chase enables small sparse networks that can be efficiently accelerated on commodity GPUs. Experimental results show that Chase achieves a significant 1.7× inference throughput speedup without sacrificing accuracy when applied to ResNet-50 on ImageNet. The authors plan to release their code to facilitate further research and adoption of their approach.

**Strengths:**

The paper exhibits exceptional clarity and a well-structured format. I particularly appreciate the bullet-point summary in the introduction, which effectively outlines the paper's roadmap. However, it must be noted that the paper can be somewhat dense and requires some additional time to fully comprehend.

The proposed method is technically sound and yields commendable empirical results. The authors offer ample implementation details that would greatly facilitate other researchers and practitioners in reproducing the key findings outlined in the paper. It is worth mentioning that the authors have also committed to releasing the code upon the acceptance of the paper, further enhancing its accessibility and potential for future advancements.

**Weaknesses:**

The paper includes experimental results that demonstrate the improvement in inference speed. However, it would be valuable to clarify whether the proposed method also enhances the training efficiency of the model. From my understanding, the primary advantage of employing dynamic sparse training lies in improving training efficiency. Otherwise, one could simply train a dense model initially and subsequently prune it to achieve sparsity. By addressing this aspect, the authors can provide a more comprehensive understanding of the benefits of their proposed method.

This paper primarily conducts its main experiments and ablation studies using CIFAR-10, which is widely regarded as a small-scale dataset. Consequently, the results obtained from this dataset may not be entirely representative. It would significantly enhance the robustness of the findings if the authors could replicate these experiments using the larger-scale ImageNet dataset. Additionally, it would be highly advantageous if the authors could extend their validation to encompass other models and tasks, such as object detection and semantic segmentation, in order to establish the general effectiveness of their proposed method across diverse domains.

**Questions:**

Please carefully address the weaknesses listed above in the rebuttal.

**Limitations:**

The paper lacks an adequate discussion on the limitations or potential negative societal impact.

---

> ### Author Rebuttal · Authors · 2023-08-10
>
> We sincerely thank the reviewer for recognizing that our proposed method is technically sound and yielding commendable empirical results. And we are glad that the reviewer appreciates the exceptional clarity and well-structured format of our paper.
> To further address all the weaknesses pointed out by you, we would like to address them point-by-point as below.
>
> **Comment 1: . Clarify whether the proposed method also enhances the training efficiency of the model.**
>
> - Please refer to our global responses.
>
>
> **Comment 2: Replicate these experiments using the larger-scale ImageNet dataset**
>
> - Thank you for your constructive feedback. We fully agree with you that evaluating our approach on large-scale ImageNet is significant. In addition to the CIFAR-level experiments, we actually have conducted numerous experiments with ResNet-50 on ImageNet, such as  Tables 4 \& 5 in the main paper, and Figure 5 and Table 8 in the appendix. Specifically, the primary evaluation considering various sparsity levels against other strong baselines is presented in Table 4 and the corresponding inference speed-ups are shared in Table 5. Besides, we've included detailed results capturing a wider range of channel-wise sparsity (from 0.2 to 0.7) with ResNet-50/ImageNet in Figure 5 and Table 8,  in Appendix B.
>
> - Moreover, as you requested, we further added the ablation study on ImageNet with ResNet-50 to further enhance the robustness of our approach. The results are highly in line with the results of CIFAR, demonstrating the strong robustness of our approach.
> | Sparsity | 90%     | 80%     |
> |:----------|---------|---------|
> | RigL     | 73.00   | 75.10   |
> | RigL+GVCP ($S_c=0.4$) | 72.11  | 73.75  |
> | RigL+GVCP+SM ($S_c=0.4$) | 73.08 | 74.18  |
> | RigL+GVCP+SM+GE (Chase) ($S_c=0.4$) | **74.03** | **75.27** |
>
> **Comment 3: Extend the validation to encompass other models and tasks.**
>
> - In response to this valuable feedback, we have expanded our evaluation to the domain of image segmentation. Specifically, we employed the DeepLab V3+ on the PASCAL VOC 2012 dataset. Staying in line with the original paper's approach that pre-trained a dense ResNet-50  on ImageNet to derive dense feature maps via atrous convolution,  we pre-trained an 80\% sparse ResNet-50 with Chase at various channel-wise sparsity levels, choosing RigL as our most direct baseline. The results are presented below, $Sc$ indicates channel-wise sparsity:
> | Method              | M-IoU (↑)    |
> |---------------------|--------------|
> | RigL                | 0.693       |
> | Chase (Sc=0.2)      | **0.697**   |
> | Chase (Sc=0.3)      | 0.696       |
> | Chase (Sc=0.4)      | 0.693       |
> | Chase (Sc=0.5)      | 0.686       |
> | Chase (Sc=0.6)      | 0.683       |
>
> - Our results show that Chase can match RigL's performance in DeepLab V3+ segmentation using only 60\% channels. This result again provides strong evidence to support the versatility of our method.
>
>
>
> **Comment 4: Lacks an adequate discussion on the limitations.**
>
>
> - We appreciate the reviewer pointing this out. One limitation of our method is its dependence on Chase's gradual channel pruning, which gradually prunes channels to reach the desired channel-sparsity. As a result, the peak memory requirement remains equivalent to that of a dense network. Moving forward, we're considering the integration of one-shot channel pruning prior to training, which we anticipate could greatly enhance efficiency.
>
> - Additionally, for the sake of convenience and ease of implementation, we currently employ masks for channel pruning during training. To truly realize training speed-up, more engineering efforts are warranted, which we plan to undertake in future iterations

---

> > ### Comment · Reviewer_bsGW · 2023-08-11
> >
> > Thank you for the rebuttal. Most of my concerns have been addressed. I think the paper is ready for publication!

---

> > > ### Author Response · Authors · 2023-08-11
> > > **Thanks for your support**
> > >
> > > Dear Reviewer bsGW,
> > >
> > > We wish to express our sincere gratitude for your valuable insights and comments. Your support is greatly appreciated!
> > >
> > > Warm regards,
> > >
> > > The Authors

---

### Author Rebuttal · Authors · 2023-08-10




We express our gratitude to all reviewers for their insights and will endeavor to address each comment separately.  The reviewers tend to have a concern regarding the training speedup, to which we aim to provide a global response:

- For the sake of convenience and easy implementation, we adopt masks
to perform channel pruning during training. Therefore, our current implementation does not have training speedups. During inference, we reshape the network and completely remove all the channels that are masked out during training, to obtain the real inference speed-ups.  We believe that enabling real inference speed-up of dynamic sparse training (DST) itself is a solid contribution as the existing DST approach usually yields unstructured sparsity which can not be directly accelerated on common hardware.

- However, our mask implementation strictly follows the real channel pruning setting, that is, when the output channels in
the current layer are removed, the input channels for the next convolutional layer as well as the BatchNorm layers are also masked. In addition, in contrast to soft channel pruning [1], our channel pruning adopts hard channel pruning, i.e., weights of masked channels are completely zeroed-out and receive no gradient updates
during the backward passes. Therefore, we are confident that our approach also holds the promise to achieve real training speed-ups and we just need more engineering effort. We will leave it for future work.

- To help better understand the benefits of training speedup, we measure the reduction of training FLOPs **solely caused by channel pruning**, without considering any unstructured sparsity. In this context, channels with unstructured sparsity are deemed dense. This evaluation mirrors computational costs on common GPUs, thereby providing a true representation of actual training expenses. With a 40\% channel sparsity, Chase is able to achieve a 1.4$\times$ training speedup when training ResNet-50 on ImageNet for 100 epochs. We expect to receive a higher speedup if we consider the benefits of unstructured sparsity together, which is widely accepted by the community.

[1] He, Yang, et al. "Soft filter pruning for accelerating deep convolutional neural networks." arXiv preprint arXiv:1808.06866 (2018).

Lastly, we again thank all the reviewers and hope our additional results with rebuttal responses can clarify their doubts. We are more than happy to provide any further explanations required.

Sincerely yours

Authors

---

### Decision · Program_Chairs · 2023-09-21

**Decision:**

Accept (poster)

**Comment:**

The paper makes a great case in favor of channel-wise sparsity andthe AC and reviewers all feel that the additional experiments and constructive discussion will greatly strengthen the submission. We therefore urge the authors to incorporate these materials. We also appreciate the honesty of the authors regarding training speedup. Please also include your points on this topic in the manuscript.